# Viral Metagenomics Reveals Diverse Viruses in Tissue Samples of Diseased Pigs

**DOI:** 10.3390/v14092048

**Published:** 2022-09-15

**Authors:** Shixing Yang, Dianqi Zhang, Zexuan Ji, Yuyang Zhang, Yan Wang, Xu Chen, Yumin He, Xiang Lu, Rong Li, Yufei Guo, Quan Shen, Likai Ji, Xiaochun Wang, Yu Li, Wen Zhang

**Affiliations:** 1School of Medicine, Jiangsu University, Zhenjiang 212013, China; 2College of Animal Sciences and Techologies, Anhui Agricultural University, Hefei 230036, China

**Keywords:** viral metagenomics, tissue samples, porcine viral diseases, eukaryotic viruses, virus evolution

## Abstract

The swine industry plays an essential role in agricultural production in China. Diseases, especially viral diseases, affect the development of the pig industry and threaten human health. However, at present, the tissue virome of diseased pigs has rarely been studied. Using the unbiased viral metagenomic approach, we investigated the tissue virome in sick pigs (respiratory symptoms, reproductive disorders, high fever, diarrhea, weight loss, acute death and neurological symptoms) collected from farms of Anhui, Jiangsu and Sichuan Province, China. The eukaryotic viruses identified belonged to the families *Anelloviridae*, *Arteriviridae*, *Astroviridae*, *Flaviviridae*, *Circoviridae* and *Parvoviridae*; prokaryotic virus families including *Siphoviridae*, *Myoviridae* and *Podoviridae* occupied a large proportion in some samples. This study provides valuable information for understanding the tissue virome in sick pigs and for the monitoring, preventing, and treating of viral diseases in pigs.

## 1. Introduction

China is one of the biggest pig-breeding countries in the world. According to the Ministry of Agriculture and Rural Affairs of the People’s Republic of China’s official statistics, the number of pigs slaughtered in 2021 was 1.672 billion (http://www.moa.gov.cn/)(accessed on 20 June 2021). In China, the traditional free-range mode of pig breeding has been changing to intensive and industrialized pig farming. Highly intensive breeding poses a significant challenge for controlling and preventing pig diseases, especially viral ones which are still the main threat to the pig industry. Domesticated pigs act as reservoirs for many emerging and re-emerging viruses (e.g., classical swine fever virus, pseudorabies virus, swine influenza virus, porcine reproductive and respiratory syndrome virus, porcine circovirus and porcine parvovirus) [1,2,3]. Some are zoonotic viruses that recognize pigs as hosts and are transmitted to humans (e.g., hepatitis E, Nipah, influenza A and Japanese encephalitis viruses) [4,5,6,7]. To ensure the food safety of pork and minimize the severe consequences of pig-associated zoonotic viruses to human health, the study of pig viromes is of critical importance.

Viral metagenomics is a powerful tool for exploring new and existing viruses, and is used to elucidate the pig virome of various samples [8,9]. However, little is known about the virome in tissue samples of diseased pigs in China.

In this study, we described the virome in tissue samples from diseased pigs collected on farms in China’s Anhui, Jiangsu and Sichuan provinces. The results present a landscape of the virome in diseased pigs and provide valuable information for the prevention and treatment of pig viral diseases.

## 2. Materials and Methods

### 2.1. Sample Collection and Preparation

From 2017 to 2018, 90 stillborn or sick pigs of different ages (suckling, nursery, and finishing), from 90 pig farms which have more than 100 pigs in stock, were sent to Anhui Agricultural University for pathogen detection (81 pigs from Anhui province, 6 from Jiangsu province, and 3 from Sichuan province) (Appendix A and Appendix A). The clinical symptoms of sick pigs included respiratory symptoms (28.9%), reproductive disorders (11.1%), high fever (3.3%), diarrhea (41.1%), weight loss (8.9%), acute death (5.6%) and neurological symptoms (2.2%)**.** The experiment was approved by the Jiangsu University and Anhui Agricultural University Ethics Committee on the use of animals and complied with Chinese ethics laws and regulations. Sick pigs were slaughtered after death by electric shock, and depending on the clinical symptoms of different pigs, various tissues (i.e., liver, lung, lymph node, spleen, kidney and brain) were aseptically collected. Tissue samples were sent to the laboratory on dry ice and stored at −80 °C before use.

### 2.2. Viral Nucleic Acid Extraction

Each tissue sample (~25 mg) was added to 800 µL phosphate-buffered saline (PBS) and homogenized in a sterilized centrifugal tube using a tissue homogenizer (Bionoon-24LD from Shanghai Bionoon Biotechnology Co., Ltd., Shanghai, China). The tissue homogenate was frozen in a −80 °C refrigerator, thawed three times on ice and centrifuged (10 min, 15,000× *g*). The supernatant was collected. In total, 500 µL tissue suspension was filtered through a 0.45 μm filter (Merck Millipore, MA, USA), to remove bacterial and eukaryotic cell-sized particles. Samples were treated with DNase (Turbo DNase from Ambion, Thermo Fisher, Waltham, MA, USA; Baseline-ZERO from Epicentre, Charlotte, USA) and RNase (Promega, Madison, WI, USA), to digest unprotected nucleic acid at 37 °C for 90 min. Viral RNA and DNA were extracted using the QIAamp Viral RNA Minikit (Qiagen, HQ, Germany).

### 2.3. Library Construction and Bioinformatics Analysis

The ninety viral-nucleic-acid pools containing DNA and RNA viral sequences were subjected to RT reactions with SuperScript III Reverse Transcriptase (Invitrogen, CA, USA) using 100 pmol of a random hexamer primer. Klenow enzyme (New England Biolabs, Ipswich, MA, USA) was used to generate the complementary chain of cDNA. Ninety libraries were constructed using Nextera XT DNA Sample Preparation Kit (Illumina, CA, USA) and sequenced using the Miseq Illumina platform with 250-base paired-ends with a distinct dual-molecular tag for each pool.

For bioinformatics analysis, paired-end reads of 250 bp generated by MiSeq were debarcoded using vendor software from Illumina. An in-house analysis pipeline running on a 32-node Linux cluster was used to process the data. The sequence reads were considered duplicates if bases 5 to 55 were identical; only one random copy of duplicates was kept. Clonal reads were removed and low-sequencing-quality tails were trimmed using Phred quality score with ten as the threshold. Adaptors were trimmed using the default parameters of VecScreen, an NCBI BLASTn with specialized parameters designed for adaptor removal. Bacterial reads were subtracted by mapping the bacterial nucleotide sequences from the BLAST NT database using Bowtie2 v2.2.4. The cleaned reads were de-novo assembled by SOAPdenovo2 version r240 using Kmer size 63 with default settings [10]. The assembled contigs and singlets were aligned to an in-house viral proteome database using BLASTx (v.2.2.7) with an E-value cut-off of < 10^−5^. The database was compiled using the NCBI virus reference proteome (https://ftp.ncbi.nih.gov/refseq/release/viral/) (accessed on 20 December 2021) and added viral protein sequences from the NCBI NR fasta file (based on annotation taxonomy in Virus Kingdom). The candidate viral hits were compared to an in-house non-virus non-redundant (NVNR) protein database to remove false positive viral hits. The NVNR database was compiled using non-viral protein sequences extracted from the NCBI NR fasta file (based on annotation taxonomy excluding the Virus Kingdom). To obtain the complete genomes or longer contigs, each viral contig was used as a reference for mapping the raw data using the Low Sensitivity/Fastest parameter in Geneious v11.1.2.

### 2.4. Phylogenetic Analysis

Phylogenetic analysis was performed based on the predicted amino acid sequences, the closest viral relatives based on the best BLASTx hit, and the representative members of related viral species or genera. Sequence alignment was performed using Clustal W with the default settings [11]. Aligned sequences were trimmed to match the genomic regions of the viral sequences obtained in the study. A phylogenetic tree with 500 bootstrap resamples of the alignment data sets was generated using the neighbor-joining method based on the *p*-distances model in MEGA-X [12]. Bootstrap values (based on 500 replicates) for each node are given. Putative ORFs in the genome were predicted by combining Geneious 11.1.2 software and the NCBI ORF finder.

### 2.5. Nucleotide Sequence Accession Number

The viral genome sequences were deposited in the GenBank with the accession numbers: MW853923 to MW853962. The raw sequence reads from the metagenomic library were deposited in the Shirt Read Archive of the GenBank database (Appendix A).

## 3. Results

### 3.1. Viral Metagenomic Overview

After next-generation sequencing with the Illumina Miseq platform, the 90 libraries generated 16,715,303 raw sequence reads. A total of 1,884,060 unique sequence reads with an E-value cut-off of <10^−5^ with viral proteins (Appendix A) corresponded to 11.27% of the total number of unique reads. Sequences related to prokaryotic viruses accounted for 69.94% and were affiliated with three virus families: *Siphoviridae* (68.92%)*, Myoviridae* (0.76%) and *Podoviridae* (0.26%). Sequences related to eukaryotic viruses accounted for 30.06% and were affiliated with six virus families: *Parvoviridae* (24.12%), *Circoviridae* (3.97%), *Flaviviridae* (1.58%)*, Arteriviridae* (0.30%)*, Anelloviridae* (0.07%)*,* and *Astroviridae* (0.02%) (Figure 1a).

Figure 1b shows a heat map of each library’s percentage of viral reads, catalogued into nine eukaryotic or prokaryotic viral families. The viral families that cause diseases in pigs (i.e., *Parvoviridae, Circoviridae*, *Flaviviridae*, and *Arteriviridae*) were distributed in different libraries. The viral sequence reads mapped to *Parvoviridae* had the widest distribution in libraries, followed by *Circoviridae*, *Flaviviridae* and *Arteriviridae*. Only a few libraries contained viral sequence reads belonging to the family *Anelloviridae*, which has no association with pig diseases, and *Astroviridae*, which causes subclinical symptoms in infecting pigs. In some libraries (i.e., pig13, pig25, pig46, pig61, pig82, and pig90) the dominant proportion of viral reads were mapped to prokaryotic viral families of *Siphoviridae, Myoviridae*, and *Podoviridae*, which may hint bacterial diseases present in sick pigs [13].

### 3.2. Classical Swine Fever Virus

Classical swine fever virus (CSFV) was detected in 18 libraries (number of sequence reads matched to CSFV ≥ 10). Diarrhea was the main clinical symptom of CSFV-infected pigs. Nine nearly complete genomes of CSFV were obtained (AH-CSFV20178-1: 12,068 bp, AH-CSFV20178-2: 12,079 bp, AH-CSFV20178-3: 12,109 bp, AH-CSFV20178-4: 12,127 bp, AH-CSFV20178-5: 12,157 bp, AH-CSFV20178-6: 12,182 bp, AH-CSFV20178-7: 12,189 bp, AH-CSFV20178-8: 12,200 bp, AH-CSFV20178-9: 12,363 bp), which had only one open reading frame (ORF) encoding 3898 amino acids (aa) of polyprotein. The aa homology of polyprotein among the eight strains (AH-CSFV20178-2 to AH-CSFV20178-9) was 99.6%, while a lower aa homology (96.6%) was obtained in the strain AH-CSFV20178-1 and the other eight strains. Twelve viral proteins were predicted by comparing the submitted CSFV sequences (GenBank NC_002657), including four structural proteins (capsid protein, RNase protein, E1 and E2) and eight non-structural proteins (N^pro^, P7, NS2, NS3, NS4A, NS4B, NS5A and NS5B) (Figure 2a). Blastn search in NCBI based on the complete genome of the nine viruses showed that eight strains (AH-CSFV20178-2 to AH-CSFV20178-9) shared the highest homology (98.98%–99.16%) with strain BJ2-2017 (GenBank MG387218), detected in Beijing (China) in 2017. The strain AH-CSFV20178-1 presented high homology (97.38%) with the strain GD53/2011 (GenBank KP343640), detected in Jilin (China) in 2011.

The phylogenetic trees for the encoding sequences of E2 indicated that the nine viruses belong to two distinct sub-genotypes (Figure 2b). The eight strains (AH-CSFV20178-2 to AH-CSFV20178-9) clustered with strains isolated from China, Japan, Mongolia, Korea, Vietnam and Lithuania formed a separate clade and belonged to sub-genotype 2.1. In terms of genetic distance, those eight viruses were phylogenetically more related to the strain BJ2-2017 isolated from Beijing, China in 2017, while they had a relatively distant relationship with the Elsenburg strain isolated from South Africa in 2005. The strain AH-CSFV20178-1 clustered with other strains of CSFVs detected from China and Vietnam formed a separate clade and belonged to sub-genotype 2.5. AH-CSFV20178-1 was phylogenetically more related to the strain GXRX2-2018 isolated from Guangdong, China in 2018.

### 3.3. Porcine Reproductive and Respiratory Syndrome Virus

In this study, porcine reproductive and respiratory syndrome virus (PRRSV) was identified in 15 libraries (number of sequence reads matched to PRRSV ≥ 10). Pigs infected with PRRSV showed diarrhea and respiratory symptoms. One nearly complete genome was assembled from library pig28: AH-PRRS20178-1. The AH-PRRS20178-1 genome was 14,940 bp in length, including a 144 bp 5’ end and a 121 bp 3’ end sequence (Figure 3a), ten ORFs (ORF1a, ORF1b, ORF2a, ORF2b, ORF3, ORF4, ORF5a, ORF5, ORF6 and ORF7) and shares a homology of 94.05% with the strain NADC30 (GenBank JN654459) isolated from the USA in 2008. This indicated that the AH-PRRS20178-1 strain belongs to type-2 PRRSV. Each ORF from the AH-PRRS20178-1 genome was compared with the strain NADC30. The results showed that AH-PRRS20178-1 shared a homology of 93.51–97.30% with the strain NADC30. In comparison with the strains NADC30 and VR-2332, two deletions were identified in the ORF1a of AH-PRRS20178-1 and the train NADC30. The first deletion in the nsp2 coding region at position 322–432 was 111 aa long. The second deletion was also found in the nsp2 coding region, at positions 504–522, and was 19 aa long. In addition, AH-PRRS20178-1 possessed one unique aa deletion in the nsp2 coding region at position 483, different from the strain NADC30 (data not shown).

The phylogenetic tree was constructed based on the ORF5 coding sequence of PRRSVs (Figure 3b) and showed that AH-PRRS20178-1 clustered with other strains detected in China and USA formed a clade and belonged to lineage 1 of type-2 PRRSV.

### 3.4. Porcine Circovirus Type 2

In the present study, porcine circovirus type 2 (PCV2) was detected in 18 libraries (number of sequence reads matched to PCV2 ≥ 10). Pigs infecting PCV2 mainly showed respiratory symptoms. Ten complete genomes of PCV2 were obtained from 10 different libraries and were named AH-PCV20178-1 to AH-PCV20178-10. All ten complete PCV2 genomes were 1767 bp in length and had four ORFs (except for AH-PCV20178-1, which missed an ORF4 because of one nucleotide mutation at the start codon) (Appendix A). ORF1 encodes a 314 aa Rep protein involved in viral replication, ORF2 encodes a 233 or 234 aa capsid protein, ORF3 encodes a 104 aa protein involved in viral pathogenesis and ORF4 encodes a 59 aa protein associated with apoptosis suppression.

The aa identity of Rep proteins among the ten isolates (AH-PCV20178-1 to AH-PCV20178-10) was 99.0~100% (Figure 4a). AH-PCV20178-3 had a shared aa identity of 100% with AH-PCV20178-4 and AH-PCV20178-6 to AH-PCV20178-10. AH-PCV20178-1 had the highest aa identity of 99.7% to AH-PCV20178-5, while AH-PCV20178-2 shared aa identity of 99.7% to seven strains except for the strains AH-PCV20178-1 (99.0% aa identity) and AH-PCV20178-5 (99.4% aa identity). Sequence comparison based on Rep proteins of all ten strains showed high homology among them; only a few amino acid sites were different (Appendix A).

The aa identity of capsid proteins among the ten isolates ranged between 94.0% and 100% (Figure 4b). AH-PCV20178-1 shared the highest aa identity of 100% to AH-PCV20178-5 based on capsid proteins. AH-PCV20178-2, AH-PCV20178-9, AH-PCV20178-3 and AH-PCV20178-10 had 100% aa identity. AH-PCV20178-6, AH-PCV20178-7 and AH-PCV20178-8 had 100% aa identity. AH-PCV20178-4 had a 99.6% aa identity to AH-PCV20178-6, AH-PCV20178-7 and AH-PCV20178-8. Amino acid sequence comparison based on capsid proteins of all ten strains showed that four of the ten isolates had a typical “SNPLTV” motif, present in most PCV2d, and the rest had a typical “SNPRSV” motif, present in PCV2b (Figure 5). The aa sequences of AH-PCV20178-1 to AH-PCV20178-3 and AH-PCV20178-5 were identical except for site 169, but 11 aa sites were different with the rest of six strains. In addition, an extended lysine residue presented at the C-terminal of AH-PCV20178-1 to AH-PCV20178-3 and AH-PCV20178-5, not found in the other six isolates. It is noteworthy that AH-PCV20178-9 and AH-PCV20178-10 had three sites different from the remaining eight isolates, while two sites were identical to the PCV2d reference strain (GenBank FJ644927) isolated from Zhejiang Province, China. Using the complete capsid encoding sequences of these, their closest viral relatives are based on the best BLASTx hits and other representative members of PCV2s. Results from the phylogenetic tree indicated that the ten viruses fell into two distinct genotypes, including PCV2b and PCV2d (Figure 4c). Six strains (AH-PCV20178-4 and AH-PCV20178-6 to AH-PCV20178-10) belonged to the PCV2b genotype, clustered with strains detected in China, United Kingdom, Slovakia, South Korea, South Africa and Namibia. Four strains (AH-PCV20178-1 to AH-PCV20178-3 and AH-PCV20178-5) belonged to the PCV2d genotype, clustered with one strain (GenBank FJ644927) isolated from Zhejiang, China.

### 3.5. Porcine Parvovirus

Porcine parvoviruses (PPV) were detected in 39 libraries. Pigs infected with PPVs showed multiple symptoms including respiratory symptoms, reproductive disorders, high fever, diarrhea, weight loss, acute death and neurological symptoms. We obtained 20 nearly complete genomes from 17 libraries. Four of the viruses belonged to PPV6 (AH-PPV620178-1 to AH-PPV620178-4; ranging from 6287 to 6571 bp), three to PPV5 (AH-PPV520178-1 to AH-PPV520178-3; ranging from 5480 to 5760 bp), seven to PPV2 (AH-PPV220178-1 to AH-PPV220178-7; ranging from 5381 to 5754 bp), five to PPV7 (AH-PPV720178-1 to AH-PPV720178-5; ranging from 4003 to 4331 bp) and one to PPV3 (AH-PPV320178; 4966 bp).

The genomes encoded two ORFs, except for AH-PPV320178, which has three ORFs (Appendix A). The ORF1 encoded a nonstructural protein (NS) with a length of 661 aa for PPV2 (except for AH-PPV220178-7 that was missing 12 aa at site 217 to 228), 636 aa for PPV3, 601 aa for PPV5, 662 aa for PPV6 and 672 aa for PPV7. The ORF2 encoded a structural protein (VP) with a length of 1032 aa for PPV2, 925 aa for PPV3, 991 aa for PPV5, 1189 aa for PPV6 and 474 aa for PPV7, except for AH-PPV720178-4 which missed five aa at site 181 to 185. In addition, PPV3 had an ORF3 which encoded a 555 aa major structural protein that is completely included in ORF2. The aa identity of NS was 96.4% to 99.5% among the PPV2 strains, 99.8% to 100% among the PPV5 strains, 100% among the PPV6 strains and 93.3% to 99.6% among the PPV7 strains. The aa identity of VP was 94.8% to 99.7% among the PPV2 strains, 98.9% to 99.6% among the PPV5 strains, 96.6% to 99.3% among the PPV6 strains and 91.5% to 100% among the PPV7 strains (Figure 6). The only PPV3 strain identified shared a homology of 99.47% to the strain HK5 (GenBank EU200675) isolated from Hong Kong (China). The phylogenetic tree constructed based on the NS1 sequences indicated that the 20 viruses belonged to five different genotypes: PPV2, PPV3, PPV5, PPV6 and PPV7 (Figure 7). Among them, seven isolates belonged to PPV2 and, clustered with a strain isolated from China, formed a clade. One isolate clustered with the HK5 strain isolate from China to form a clade of PPV3. Three isolates belonged to PPV5 and clustered with strains isolated from South Korea, Poland and the United States to form a clade. Four isolates belonged to PPV6 and clustered with strains isolated from China, Brazil, Poland and South Korea to form a clade. Five PPV7 isolates clustered with a strain isolated from China and Colombia to form a clade.

### 3.6. Other Viruses That Only Obtained Short Fragments

Apart from described viruses, some anelloviruses and astroviruses were also identified in the pig tissue samples. Anelloviruses were present in six libraries, and these sequences showed high sequence identity (95%) to other Torque teno sus viruses (TTSuV). Astroviruses were present in four libraries, and these sequences showed high sequence identity (99%) to the strain PoAstV_VIRES_GZ01_C1 detected in Jilin Province (China). Because no large contig was obtained using the de novo assemble program in Geneious, phylogenetic analyses for anellovirus and astrovirus were not performed.

## 4. Discussion

A variety of viruses can infect pigs and cause diseases. Zoonotic viruses from pigs can be transmitted to humans, so monitoring viruses in pigs is essential for ensuring the development of the pig industry while protecting human health. In previous studies, viral metagenomics has been applied to searching for novel viruses in various samples from pigs. These novel viruses include bocaviruses, Torque teno viruses, astroviruses, rotaviruses and kobuviruses [14]. Viral metagenomics has also been widely used to explore the etiology of porcine diseases (i.e., piglet diarrhea, postweaning multisystemic wasting syndrome, porcine respiratory diseases, periweaning failure-to-thrive syndrome (PFTS)) [15,16]. Only a few viral metagenome studies have explored the etiology of porcine diseases using pig tissues. Mikael and colleagues [9] analyzed the virus composition of tonsils from conventional pigs with lesions in the respiratory system and from specific pathogen-free pigs. The authors testified that no differences were observed among the different groups. Giovanni and colleagues [16] investigated the potential role of viral agents in PFTS-infected and healthy pigs, evaluating the virome composition of different organs. These authors demonstrated a higher abundance of porcine parvovirus 6 in healthy pigs. Contrarily, ungulate bocaparvovirus 2, ungulate protoparvovirus 1 and porcine circovirus 3 abundance was higher in pigs with PFTS [9,16]. The viral metagenomic research on the correlation between viruses and porcine diseases needs to be strengthened.

In the present study, we performed high-throughput sequencing to uncover the virome of various tissues collected from diseased pigs. No zoonotic viruses were detected in the tissue libraries, however, many viruses associated with pig disease were detected, including CSFV, PRRSV, PCV and PPV. These viruses are widespread in the pig farms of China and cause significant economic losses to the pig industry.

The classical swine fever virus belongs to the genus *Pestivirus* within the family *Flaviviridae*. The viral genome is a single-stranded, positive-sense RNA genome of approximately 12.3 kb with one ORF flanked by 5’ and 3’ untranslated regions (UTRs). The ORF encodes a 3898 aa polyprotein, later cleaved into four structural proteins (C, E^rns^, E1 and E2) and eight non-structural proteins (N^pro^, P7, NS2, NS3, NS4A, NS4B, NS5A and NS5B). More recently, based on the phylogenetic analysis of the sequences of 53 complete E2 envelope glycoprotein genes, CSFVs were divided into five genotypes (1 to 5) and 17 subgenotypes (1.1–1.7, 2.1–2.7, 3, 4 and 5) [17]. In China, four subgenotypes (1.1, 2.1, 2.2 and 2.3) of CSFV strains have been identified in mainland China and have contributed to CSFV outbreaks. Among them, subgenotype 2.1 isolates, especially 2.1b, have become predominant in the last decade and endemic in many regions of China [18]. This study obtained nine complete genomes of CSFV from different porcine tissue samples. The phylogenetic analysis based on the E2 gene showed that eight isolates belonged to subgenotype 2.1, phylogenetically related to the BJ2-2017 strain isolated from Beijing (China). Another strain was assigned to subgenotype 2.5 and phylogenetically related to the GXRX2-2018 strain isolated from Guangdong (China) in 2018. The result indicated that two different subgenotypes of CSFV were endemic to this region. These results agree with previous studies that reported the subgenotype 2.1 as predominant in China.

Porcine reproductive and respiratory syndrome, characterized by reproductive failure in sows and respiratory disease in pigs of all ages, is one of the most devastating swine diseases worldwide [19]. The etiologic agent is the porcine reproductive and respiratory syndrome virus that belongs to the family *Arteriviridae*, order *Nidovirales*. This is a kind of enveloped, positive single-stranded RNA virus. The genome of PRRSV is approximately 15 kbp, and has at least 10 ORFs flanked by 5’ and 3’ UTRs. ORF1a and ORF1b encode viral non-structural proteins, while ORF2a, ORF2b and ORF3-7 encode viral structural proteins [20]. PRRSV is divided into European species 1 and North American species 2, with Lelystad and VR-2332 as prototypical strains [21]. In China, the coexistence of the two species has been previously identified, but species 2 of PRRSV is the predominant strain [22]. In recent years, NADC30-like PRRSV has been identified in many regions of China and has caused substantial economic losses to the porcine industry. The NADC30-like PRRSV is genetically similar to the NADC30 strain, a type-2 PRRSV isolated in the United States in 2008 [23]. This study identified PRRSV in 15 porcine tissue libraries, and one nearly complete genome of PRRSV (AH-PRRS20178-1) was obtained. The AH-PRRS20178-1 strain shared the highest homology of 94.05% with the strain NADC30. Similar to the previously reported NADC30-like PRRSVs, the AH-PRRS20178-1 has 131 aa discontinuous deletions in the nsp2, including a 111 aa deletion at position 322–432, 1 aa deletion at position 483 and a 19 aa deletion at position 504–522, which could distinguish them from other PRRSV strains. This is the first identification of NADC30-like PRRSV in Anhui Province, China. The nsp2 gene of PRRSV is highly variable and includes naturally occurring mutations, insertions and deletions, which might be the most important marker for monitoring the genetic variation and evolution of PRRSV [24]. Similar deletions have been detected in a previous study by Zhou and colleagues that reported a deletion in nsp2 was associated with milder virulence of the HP-PRRSV strains. However, they concluded that the 131 aa deletion was not related to the virulence of PRRSV in China [25].

In this study, the clinical symptoms of pigs infected with NADC30-like PRRSV were diarrhea, respiratory symptoms and high fever. Although NADC30-like PRRSV is not pathogenic alike other highly pathogenic PRRSVs, it is distinguished by a high incidence of recombination with other virus strains, which might lead to a virulence change. These characteristics probably made current vaccines ineffective and confers the viruses with a far greater ease with which to escape immune surveillance. Therefore, monitoring of the mutation sites of NADC30-like PRRSV should be strengthened.

The porcine circoviruses (PCVs), the least-known animal viruses, belong to the genus *Circovirus* of the family *Circoviridae*. Nowadays, four PCVs have been identified, namely, porcine circovirus 1 (PCV1), porcine circovirus 2 (PCV2), porcine circovirus 3 (PCV3) and porcine circovirus 4 (PCV4) [26]. The present study identified PCV2 in ten libraries. The positive rate of PCV2 in this study was 20.0% (18/90) and is higher that reported by Canal and colleagues, who identified a 2.16% positive rate for PCV2 in porcine liver samples from slaughtered swine in the state of Rio Grande do Sul, Brazil [27]. A recent survey on the prevalence of PCV2 in China from 2015 to 2019 showed that PCV2 was widely distributed throughout China. The average prevalence of PCV2 infection was 46.0% in China. The prevalence of PCV2 infection in China was 32.3% before 2015, 42.3% between 2015 and 2017 and 51.9% in 2017 to 2019 [28]. The study’s prevalence of PCV2 infection was lower than the average percentage in China. The prevalence of PCV2 infection may be caused by differences in sampling type or management measures implemented in each pig farm. PCV2 causes postweaning multisystemic wasting syndrome, porcine respiratory disease complex, reproductive disease, porcine dermatitis and nephropathy and enteritis. Pigs infected with PCV2 in this study presented respiratory symptoms, diarrhea, high fever, neurological symptoms and reproductive disorders. Based on the ORF2 gene sequences, PCV2 is divided into eight genotypes (a to h). PCV-2a, PCV-2b and PCV-2d are widespread and similarly virulent in pigs, while the clinical significance of the remaining genotypes is unknown [1]. Recent reports indicate that an ongoing genotype shift from PCV-2b to PCV-2d is occurring, and PCV-2d has become the main epidemic strain. PCV-2d was initially called a mutant of PCV-2b and linked to potential vaccination failure cases [29]. We identified two genotypes of PCV2 here, including PCV-2b and PCV-2d. Comparing PCV-2b with PCV-2d, we found multiple aa sites of capsid protein changes, especially a lysine residue at the C-terminal extension of the capsid protein of PCV-2d. Recent experimental infection studies indicated that strains with a lysine residue at the C-terminal extension of the capsid protein led to increased virulence in vivo [30]. Based on the present study, we cannot confirm if PCV-2d has higher virulence than PCV-2b. Unlike the high variation of capsid proteins, only a few aa sites of Rep protein were different. PCV2 generally has four open reading frames (ORFs). The proteins encoded by ORF1 to ORF3 are, respectively, involved in viral replication, the immunogenic capsid protein and the viral-pathogenesis-associated protein. The ORF4 protein, as a novel discovered viral protein, induces host cell apoptosis [31]. In this study, ORF4 is missing in the genome of the strain AH-PCV20178-1 because of the mutation of the third initiation codon from “G” to “A”. Further studies need to check if the mutation can reduce the virulence of the strain AH-PCV20178-1.

Members of the family *Parvoviridae*, the family most commonly detected in this study, are small, non-enveloped viruses with a single-stranded DNA genome of 4–6 kb. *Parvoviridae* includes two subfamilies, *Parvovirinae* and *Densovirinae*, which infect vertebrates and invertebrates. The subfamily *Parvovirinae* consists of ten genera, *Amdoparvovirus*, *Artiparvovirus*, *Aveparvovirus*, *Bocaparvovirus*, *Copiparvovirus*, *Dependoparvovirus*, *Erythroparvovirus*, *Loriparvovirus*, *Protoparvovirus* and *Tetraparvovirus* [32]. Porcine parvovirus (PPV1) was a major causative agent in porcine reproductive failure, while the diseases associated with novel porcine parvoviruses (PPV2 to PPV7) have not been well-characterized [33]. The present study detected five PPV species from 39 libraries, including PPV2, PPV3, PPV5, PPV6 and PPV7. Among them, 15 (16.6%) tissue libraries positive for PPV2, 3 (3.3%) positive for PPV3, 7 (7.8%) positive for PPV5, 16 (17.8%) positive for PPV6 and 12 (13.3%) positive for PPV7 were identified. The positive rate of PPVs in this study was lower in comparison to other studies by Tomasz and colleagues that identified a positive rate of 53.9% for PPV2, 15.4% for PPV3, 19.7% for PPV5 and 24.0% for PPV6 in serum samples from Polish swine farms [33]. The difference in positive rate may be caused by regional differences or differences in sampling type. In China, a previous systematic investigation showed that PPV1-7 was highly prevalent (55.4%) in nursery and finishing pigs in recent years [34]. This study’s total PPV-positive rate was 43.3%, lower than the previous report. To our surprise, the tissue samples were collected from diseased pigs, which showed obvious clinical symptoms, including diarrhea, respiratory syndrome and reproductive disorder, but PPV1 as the sole causative agent of porcine parvovirus infection was not detected in this study. On the contrary, other PPVs were identified in many tissue samples. Although the novel PPVs (PPV2-7) were identified in 53.9% of healthy pigs in the previous study, it is not possible to rule out the possibility that they were the etiological agent of pig diseases. Further animal infection experiments need to be done to elucidate if there is any association between these novel PPVs and pig diseases. In addition, some non-pathogenic mammalian viruses such as anelloviruses and astroviruses were isolated from porcine tissue samples. Unsurprisingly, anelloviruses and astroviruses have been detected in porcine tissue samples. Mikael and colleagues [9] have previously detected those viruses in tonsil samples from conventional pigs and specific-pathogen-free pigs. Although almost one or more mammalian viruses have been identified from each library, some libraries did not present any mammalian viruses. This indicated that the cause of pig diseases in these cases was not a virus but other pathogenic microbes, such as bacteria, or even physical and chemical factors. Some libraries (pig13, pig25 and pig90) only contained prokaryotic viral sequences, indirectly indicating that bacteria were the causative agents of these pig diseases. Our hypothesis about the bacterial disease in pigs was based on discovering *Pseudomonas* and *Streptococcus* phages in these libraries. Unfortunately, bacterial cultures were not performed for pig13, pig25 and pig90. *Streptococcus suis* and pathogenic *Escherichia coli* were separately isolated from pig61 and pig82 by bacterial culture. This indicated bacteria may be the cause of pig diseases.

The application of viral metagenomics is becoming commonplace in humans’ diagnosis, prevention and control of infectious diseases [35]. It provides novel possibilities for the direct comparative analysis of virus compositions of various clinic samples and the detection of “new, emerging viruses”. An excellent example of the practical applicability of viral metagenomics is the genome sequencing of SARS-CoV-2, which provides excellent assistance in preventing and controlling COVID-19 [36]. For veterinary medicine, the application of viral metagenomics is still on the road to development. There are considerable impediments to adopting viral metagenomics for disease diagnosis in veterinary medicine, including the sensitivity of detection, turnaround time and cost. Firstly, the cost of viral metagenomics is more expensive than the traditional diagnostic methods. For small-scale pig farms, the viral metagenomic approach is not fit for large-scale epidemiological investigation and exploration of disease pathogens when considering the cost–benefit. Secondly, the turnaround time for metagenomics is measured in days rather than minutes as is the case with traditional diagnostic methods. This will significantly reduce the timeliness of diagnosis and efficiency in preventing pig diseases, especially severe infectious diseases such as African swine fever. Thirdly, the sensitivity of viral metagenomics is lower than the traditional methods. The process of viral metagenomics is complex, including sample handling, library construction, sequencing and data analysis. Error in each step will affect the quality of the library and reduce the sensitivity. In this study, we analyzed the agreement of the traditional methods (RT-PCR or RT-qPCR) and the viral-metagenomic approach in pig-disease diagnosis. The viruses (PCV2, PRRSV and CSFV) whose sequence reads ≥10 in the library were almost detected using traditional methods (RT-PCR or RT-qPCR), while the viruses in some samples that were detected using traditional methods had no sequence reads in the corresponding libraries. This indicates that the traditional methods have higher sensitivity than the viral-metagenomic approach. Based on the shortcomings of viral metagenomics, it will face a challenge in promoting the diagnosis of pig disease widely. However, through regular sampling and testing, viral metagenomics can be used to monitor the outbreak of potential diseases in pigs on those farms of a specific scale. After obtaining the associated information of the virus genome, the traditional methods will be used for further pathogen confirmation and large-scale epidemiological investigation. We believe that the combination of viral metagenomics and traditional methods in pig-disease diagnosis and prevention will have broader prospects.

## 5. Conclusions

Using the virus-metagenomics approach, we identified a variety of porcine-disease-associated viruses in tissue samples. Our study provides an overview of the tissue virome of sick pigs and significantly increases our understanding of the prevalence status of porcine-disease-associated viruses. Our study is a preliminary exploration of the application of viral metagenomics in the diagnosis of pig diseases and provides essential information for its application in the future diagnosis and prevention of pig diseases.

## Figures and Tables

**Figure 1 viruses-14-02048-f001:**
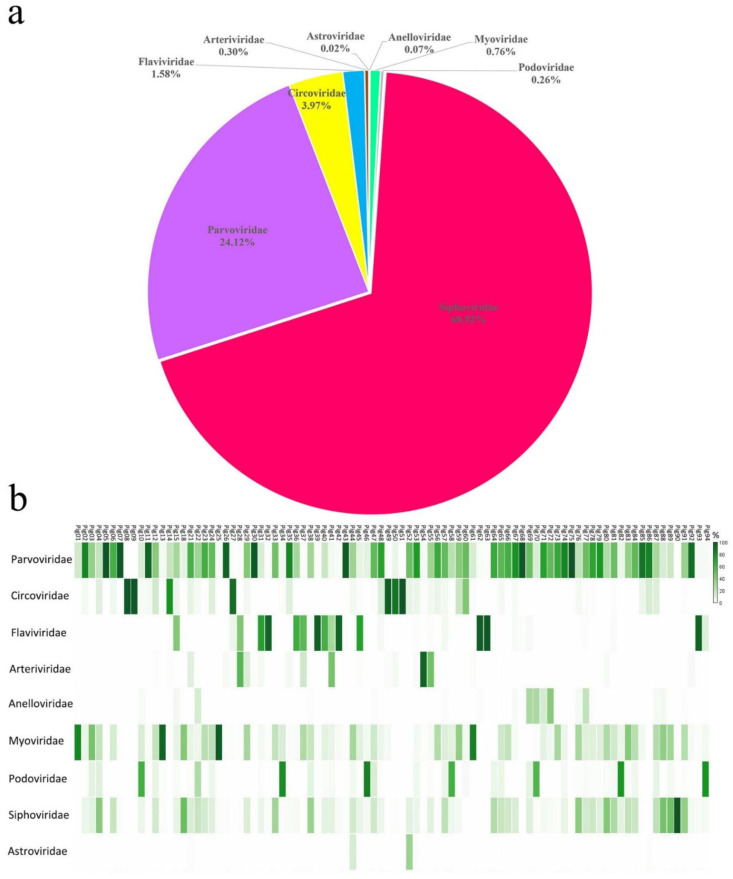
Composition of the virome from tissues collected from sick pigs: (**a**) the percentage of virus sequences from different virus families and (**b**) the percentage of eukaryotic or prokaryotic viral families in each library.

**Figure 2 viruses-14-02048-f002:**
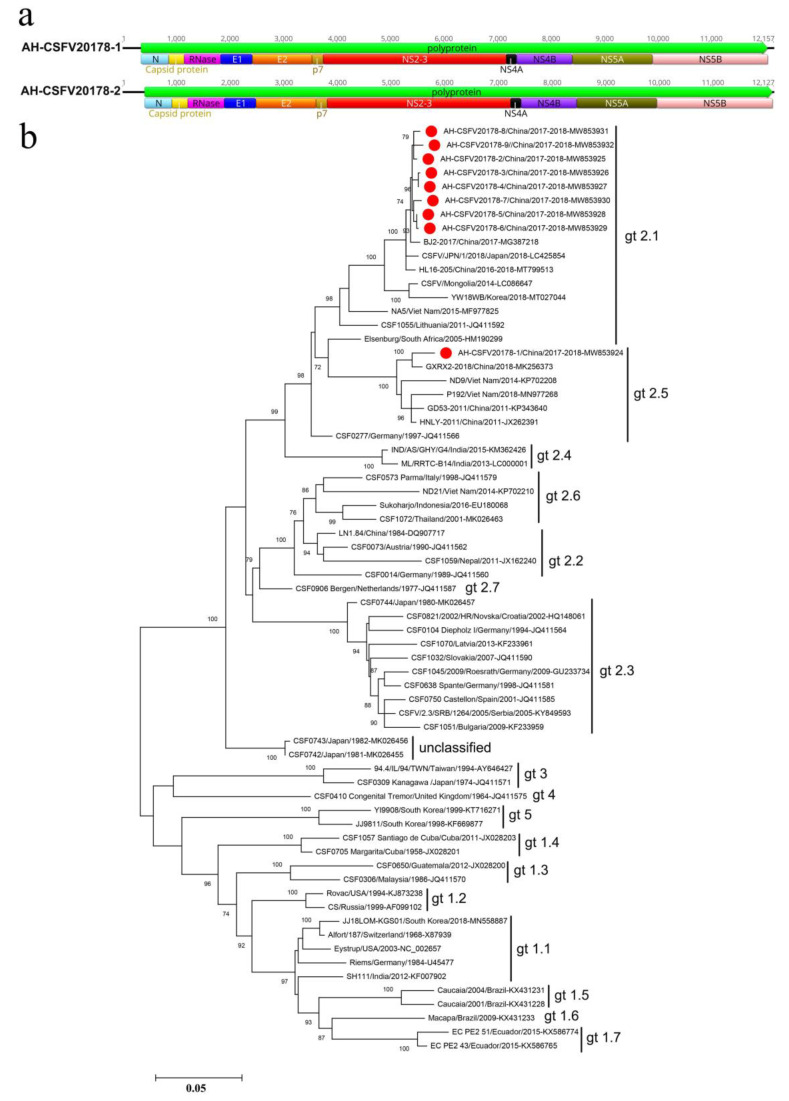
Classical swine fever virus isolated from the sick pigs: (**a**) genomic organization of two different CSFV-representative strains, including the ORFs and viral proteins encoding sequences; (**b**) phylogenetic analysis based on the sequences of E2 of CSFVs identified in this study and the reference strains of other CSFVs. CSFVs identified in this study are marked with solid red circle.

**Figure 3 viruses-14-02048-f003:**
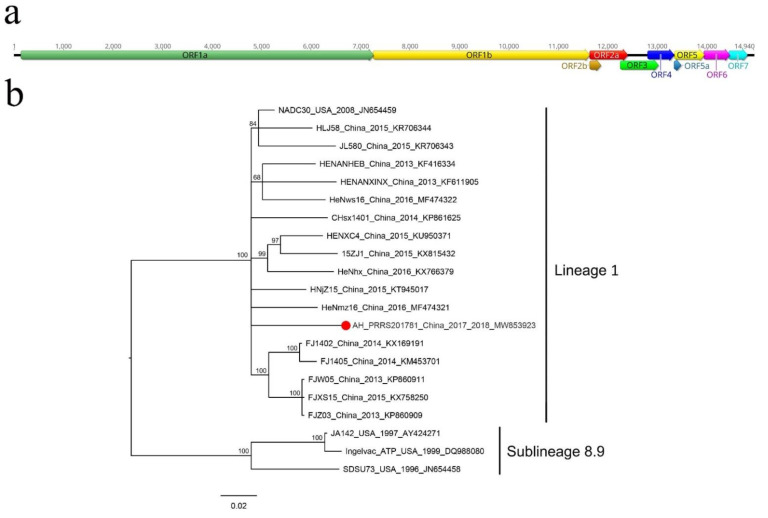
Porcine reproductive and respiratory syndrome virus isolated from sick pig: (**a**) genomic organization of AH-PRRS20178-1, including the ORFs and the viral protein encoding sequences; (**b**) phylogenetic analysis based on the ORF5 sequences of AH-PRRS20178-1 and reference strains of another type-2 PRRSV. AH-PRRS20178-1 identified in this study is marked with solid red circle.

**Figure 4 viruses-14-02048-f004:**
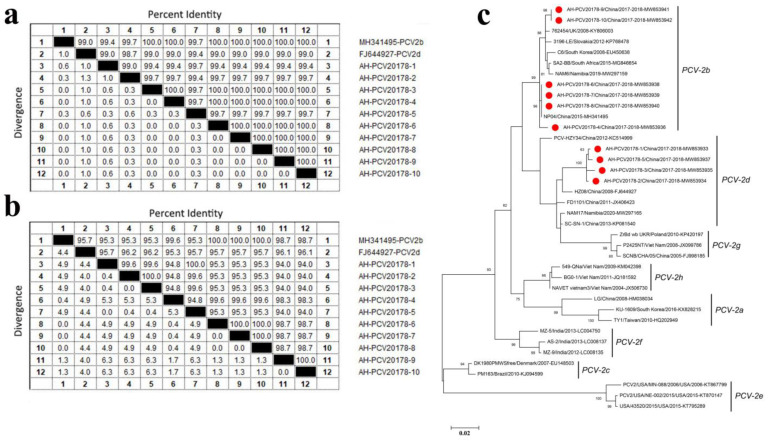
Porcine circovirus type 2 identified in sick pigs: (**a**,**b**) The sequence comparison based on the amino acid sequences of Rep or capsid protein of ten PCV2 isolates identified in this study; (**c**) phylogenetic analysis based on the nucleotide sequences of the capsid of PCV2 identified in this study and reference strains of other PCV2s. PCV2s identified in this study are marked with a solid red circle.

**Figure 5 viruses-14-02048-f005:**
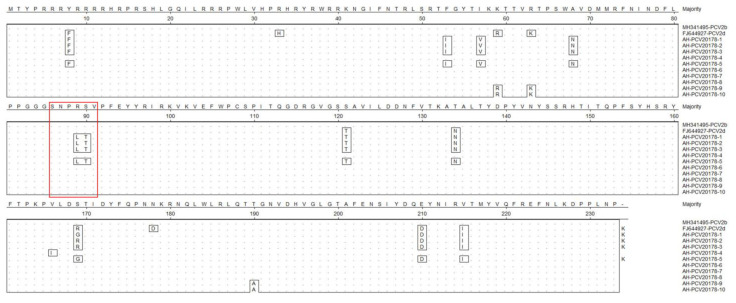
The amino acid sequence alignment of the capsid protein of PCV2 isolated from sick pigs. The differential amino acids are shown. The conservative motif “SNPRSV” was marked with a red wireframe.

**Figure 6 viruses-14-02048-f006:**
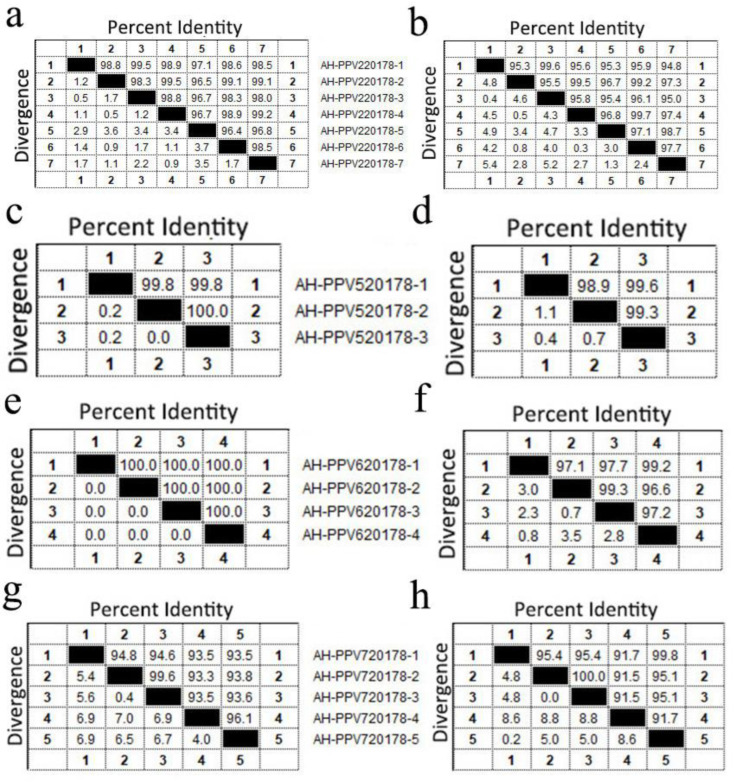
Porcine parvovirus isolated from sick pigs. Pairwise comparison based on the aa of NS protein (**a**,**c**,**e**,**g**) and VP protein of five species of PPV (**b**,**d**,**f**,**h**).

**Figure 7 viruses-14-02048-f007:**
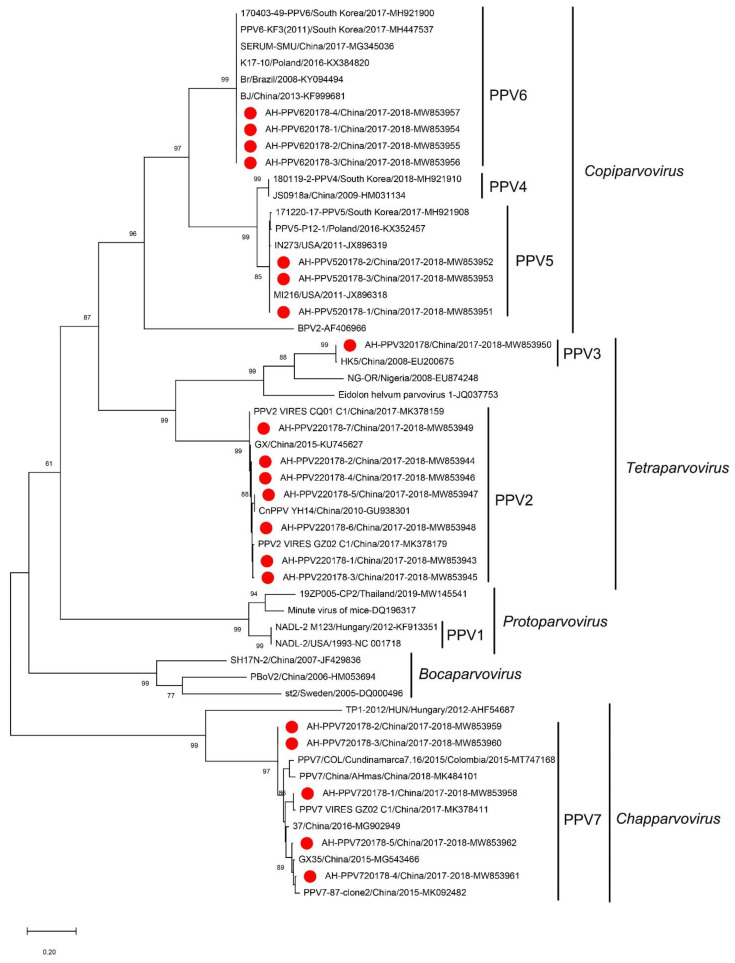
Phylogenetic analysis of PPVs isolated from sick pigs. The phylogenetic analysis is based on the NS1 sequences, including reference strains from the genus *Copiparvovirus*, *Tetrapavovirus*, *Protoparvovirus*, *Bocaparvovirus* and *Chapparvovirus*. PPVs identified in the study are marked with a solid red circle.

## Data Availability

The viral genome obtained in this study were deposited in GenBank with the accession numbers: MW853923 to MW853962. The raw sequence reads from metagenomic library were deposited in the Shirt Read Archive of GenBank database (Appendix A).

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
