# Peer review of "Viral Metagenomics Reveals Diverse Viruses in Tissue Samples of Diseased Pigs"

_viruses, 2022, doi:10.3390/v14092048_

Round 1

Reviewer 1 Report

Title: Viral metagenomics reveals diverse viruses in the tissue samples of diseased pigs

Authors: Shixing Yang, Dianqi Zhang, Zexuan Ji, Yuyang Zhang, Yan Wang, Xu Chen, Yumin He, Xiang Lu, Rong Li, Yufei Guo, Quan Shen, Likai Ji, Xiaochun Wang, Yu Li, Wen Zhang

Reference: viruses-1891787

Article type: Research

Reviewer Comments:

The manuscript viruses-1891787, entitled “Viral metagenomics reveals diverse viruses in the tissue samples of diseased pigs”, identifies the viruses that constitute the virome in tissues collected from diseased pigs.

General comments:

1. Clarity and readability of the manuscript can be improved in terms of:

a)        English language

b)       Typographical mistakes

c)        Numbers up to eleven must be written in full

d)       Consistency

e)        Units from SI (µl and not ul)

f)         For the first time it is used, an acronym should be explained. After, it is enough to use the acronym

2. Much of the text from the result section can be converted into tables, reducing the length of the manuscript and simplifying result interpretation.

Specific comments:

Line 10: Please consider replacing it with “Abstract: The swine industry plays an essential role in agricultural production in China. Diseases, es-,”.

Lines 11-12: Please consider replacing with “diseases, affect the development of the pig industry and threaten human health.”.

Lines 14 – 17: Please consider replacing it with “The eukaryotic viruses identified belonged to the families Anelloviridae, Arteriviridae, Astroviridae, Flaviviridae, Circoviridae, and Parvoviridae. The prokaryotic virus families, including Siphoviridae, Myoviridae, and Podoviridae, occupied a large proportion of some samples.”

Lines 17 – 19: Please consider replacing it with “This study provides valuable information for understanding the tissue virome in sick pigs and for the monitoring, preventing, and treating of viral diseases in pigs.”

Lines 24-27: Please consider replacing it with “China is one of the biggest pig breeding countries in the world. According to the Ministry of Agriculture and Rural Affairs of the People's Republic of China's official statistics, the number of pigs slaughtered in 2021 achieved 1.672 billion (http://www.moa.gov.cn/).”

Lines 27-28: Please consider replacing it with “With the development of the economy, the demands for a good environment and public health are increasing. ” This sentence seems quite decontextualized, please consider removing it.

Lines 28-29: Please consider replacing it with “In China, the traditional free-range pig breeding has been changing to intensive and industrialized pig farming.”

Lines 29-31: Please consider replacing it with “Highly intensive breeding poses a significant challenge for controlling and preventing pig diseases, especially viral ones, which are still the main threat to the pig industry.”

Lines 31-38: Please consider replacing it with “Domesticated pigs act as reservoirs for many emerging and re-emerging viruses (e.g., classical swine fever virus, pseudorabies virus, swine influenza virus, porcine reproductive and respiratory syndrome virus, porcine circovirus, porcine parvovirus) [1–3]. Some are zoonotic viruses that recognize pigs as hosts and are transmitted to humans (e.g., hepatitis E, Nipah, influenza A, and Japanese encephalitis viruses) [4–7]. To ensure the food safety of pork and minimize the severe consequences of pig-associated zoonotic viruses to human health, the study of pig virome is of critical importance.

Lines 39-41: Please consider replacing it with “Viral metagenomics is a powerful tool for exploring new and existing viruses used to elucidate the pig virome of various samples [8,9]. However, little is known about the virome in the tissue samples from diseased pigs in China.”

Lines 43-44: Please consider replacing it with “In this study, we described the virome in tissue samples from diseased pigs collected on farms in China's Anhui, Jiangsu, and Sichuan provinces.”

Lines 44-45: Please consider removing the sentence “Many DNA or RNA viruses associated with pig disease were identified.”

Lines 45-47: Please consider replacing it with “The results draw a landscape of virome in diseased pigs and provide valuable information for the prevention and treatment of pig viral diseases.”

Lines 50-58: Please consider replacing it with “From 2017 to 2018, 90 stillborn or sick pigs of different ages were sent to the Anhui Agricultural University for pathogen detection (81 pigs from the Anhui province, six from the Jiangsu province, and three from the Sichuan province) (Figure S1 and Table S1). The clinical symptoms of sick pigs included respiratory symptoms, reproductive disorders, high fever, diarrhea, weight loss, acute death, and neurological symptoms. Sick pigs were slaughtered, and depending on the clinical symptoms of different pigs, various tissues (i.e., liver, lung, lymph node, spleen, kidney, and brain) were aseptically collected. Tissue samples were sent to the laboratory on dry ice and stored at -80℃ before use.

Lines 60-63: Please consider replacing it with “Each tissue sample (~25 mg) was added 800 µl phosphate-buffered saline (PBS) and homogenized in a sterilized centrifugal tube using a tissue homogenizer. The tissue homogenate was frozen in a -80 ℃ refrigerator, thawed three times on ice and centrifuged (10 min, 15,000 ×g). The supernatant was collected.

Lines 63-70: Confusing sentence, please rewrite: “Totally 500 µl tissue suspensions equal from various tissues from each pig were mixed and filtered through a 0.45-μm filter (Merck Millipore, MA, USA), to remove bacterial and eukaryotic cell-sized particles. Samples were treated with DNase (Turbo DNase from Ambion, Thermo Fisher, Waltham, MA, USA; Baseline-ZERO from Epicentre, Charlotte, USA) and RNase (Promega, Madison, WI, USA), to digest unprotected nucleic acid, at 37 ℃, for 90 min. Viral RNA and DNA were extracted using the QIAamp viral RNA Minikit (Qiagen, HQ, Germany).”.

Lines 72–75: Please consider replacing it with: The ninety viral nucleic acid pools containing DNA and RNA viral sequences were subjected to RT reactions with SuperScript III reverse transcriptase (Invitrogen, CA, USA) using (?) 100 pmol of a random hexamer primer. Klenow enzyme (New England Biolabs, MA, USA) was used to generate the complementary chain of cDNA.

Lines 84-85: Please consider replacing it with: “Adaptors were trimmed using the default parameters of VecScreen, an NCBI BLASTn with specialized parameters designed for adaptor removal.”

Lines 85-87: Please consider replacing it with: “Bacterial reads were subtracted by mapping the bacterial nucleotide sequences from the BLAST NT database using Bowtie2 v2.2.4.”

Lines 88-90: Please consider replacing it with: “The assembled contigs and singlets were aligned to an in-house viral proteome database using BLASTx (v.2.2.7) with an E-value cut off of <10−5 .”

Lines 90-93: Please consider replacing it with: “This database was compiled using the NCBI virus reference proteome (https://ftp.ncbi.nih.gov/refseq/release/viral/) and added viral protein sequences from the NCBI nr fasta file (based on annotation taxonomy in the Virus Kingdom).”

Lines 93-97: Please consider replacing it with: “The candidate viral hits were compared to an in-house non-virus non-redundant (NVNR) protein database to remove false positive viral hits. The NVNR database was compiled using non-viral protein sequences extracted from the NCBI nr fasta file (based on annotation taxonomy excluding the Virus Kingdom).”

Lines 97-98: Please consider replacing it with: “To obtain complete genomes or longer contigs, each viral contig was used as a reference for mapping the raw data using the Low Sensitivity/Fastest parameter in Geneious v11.1.2.”

Lines 101-103: Please consider replacing it with: “Phylogenetic analyses were performed based on the predicted amino acid sequences, the closest viral relatives based on the best BLASTx hit, and the representative member of related viral species or genera.”

Lines 101-103: Please consider replacing it with: “The viral genome sequences were deposited in the GenBank with the accession numbers: MW853923 to MW853962. The raw sequence reads from the metagenomic library were deposited in the Shirt Read Archive of the GenBank database (Table. S2).”

Lines 116-117: Please consider replacing it with: “After next-generation sequencing with the Illumina Miseq platform, the 90 libraries generated 16,715,303 raw sequence reads.”

Lines 117-119: Please consider replacing it with: “A total of 1,884,060 unique sequence reads with an Evalue cut-off of <10−5 with viral proteins (Table S2) corresponded to 11.27% of the total number of unique reads.”

Lines 119-126: Please consider replacing it with: “Sequences related to prokaryotic viruses accounted for 69.94% and were affiliated with three virus families: Siphoviridae (68.92%), Myoviridae (0.76%), and Podoviridae (0.26%). Sequences related to eukaryotic viruses accounted for 30.06% and were affiliated with six virus families: Parvoviridae (24.12%), Circoviridae (3.97%), Flaviviridae (1.58%), Arteriviridae (0.3%), Anelloviridae (0.07%), and Astroviridae (0.02%) (Figure. 1a). ”

Lines 127-137: Please consider replacing it with: “Figure 1b shows a heat map of each library's percentage of viral reads, catalogued into nine eukaryotic or prokaryotic viral families. The viral families that cause diseases in pigs (i.e., Parvoviridae, Circoviridae, Flaviviridae, and Arteriviridae) were distributed in different libraries. The viral sequence reads mapped to Parvoviridae had the widest distribution in libraries, followed by Circoviridae, Flaviviridae, and Arteriviridae. Only a few libraries contained viral sequence reads belonging to the family Anelloviridae, which has no association with pig diseases, and Astroviridae, which cause subclinical symptoms in infected pigs. In some libraries (i.e., pig13, pig25, pig46, pig61, pig82, and pig90) the dominant proportion of viral reads were mapped to the prokaryotic viral families of Siphoviridae, Myoviridae, and Podoviridae, which may hint bacterial diseases present in sick pigs [13].”

Lines 139-141: Please consider replacing it with: “Figure 1. Composition of the virome from tissues collected from sick pigs: (a) the percentage of virus sequences from different virus families and (b) the percentage of eukaryotic or prokaryotic viral families in each library.”

Lines 143-160: Please consider replacing it with: “Classical swine fever virus (CSFV) was detected in 18 libraries (number of sequence reads matched to CSFV≥10). Nine nearly complete genomes of CSFV were obtained (AH-CSFV20178-1: 12,068 bp,  AH-CSFV20178-2: 12,079 bp, AH-CSFV20178-3: 12,109 bp, AH-CSFV20178-4: 12,127 bp, AH-CSFV20178-5: 12,157 bp, AH-CSFV20178-6: 12,182 bp, AH-CSFV20178-7: 12,189 bp, AH-CSFV20178-8: 12,200 bp, AH-CSFV20178-9: 12,363 bp), which had only one open reading frame (ORF) encoding a 3,898 amino acids (aa) of the polyprotein. The aa homology of polyprotein among the eight strains (AH-CSFV20178-2 to AH-CSFV20178-9) was 99.6%, while a lower aa homology (96.6%) was obtained in the strain AH-CSFV20178-1 and the other eight strains. Twelve viral proteins were predicted by comparing the CSFV submitted sequences (GenBank NC_002657), including four structural proteins (capsid protein, RNase protein, E1, and E2) and eight non-structural proteins (Npro, P7, NS2, NS3, NS4A, NS4B, NS5A, and NS5B) (Figure. 2a). Blastn search in NCBI based on the complete genome of the nine viruses showed that eight strains (AH-CSFV20178-2 to AH- 156 CSFV20178-9) shared the highest homology (98.98%-99.16%) with strain BJ2-2017 (GenBank MG387218) detected in Beijing (China), in 2017. The strain AH-CSFV20178-1 presented high homology (97.38%) with the strain GD53/2011 (GenBank KP343640), detected in Jilin (China), in 2011.”

Lines 161-172: Please consider replacing it with: “The phylogenetic trees for the encoding sequences of E2 indicated that the nine viruses belong to two distinct sub-genotypes (Figure. 2b). The eight strains (AH-CSFV20178-2 to AH-CSFV20178-9) clustered with strains isolated from China, Japan, Mongolia, Korea, Viet Nam, and Lithuania that formed a separate clade and belonged to sub-genotype 2.1. In terms of genetic distance, those eight viruses were phylogenetically more related to the strain BJ2-2017 isolated from Beijing of China in 2017, while they had a relatively distant relationship with the Elsenburg strain isolated from South Africa in 2005. The strain AH-CSFV20178-1 clustered with other strains of CSFVs detected from China, Viet Nam formed a separate clade and belonged to sub-genotype 2.5. AH-CSFV20178-1 were phylogenetically more related to the strain GXRX2-2018 isolated from Guangdong in China in 2018.”

Lines 174-178: Please consider replacing it with: “Figure 2. Classical swine fever virus isolated from the sick pigs: (a) genomic organization of two different CSFV representative strains, including the ORFs and viral proteins encoding sequences; (b) phylogenetic analysis based on the sequences of E2 of CSFVs identified in this study and the reference strains of other CSFVs. CSFVs identified in this study are marked with a solid red circle.”

Lines 180-200: Please consider replacing it with: “In this study, porcine reproductive and respiratory syndrome virus (PRRSV) was identified in 15 libraries (number of sequence reads matched to PRRSV≥10), and one nearly complete genome was assembled from library pig28 - AH-182 PRRS20178-1. The AH-PRRS20178-1 genome was 14,940 bp in length, including a 144-bp 5' end and a 121-bp 3' end sequences (Figure. 3a), ten ORFs (ORF1a, ORF1b, ORF2a, ORF2b, ORF3, ORF4, ORF5a, 185 ORF5, ORF6 and ORF7) and shares a homology of 94.05% with the strain NADC30 (GenBank JN654459) isolated from the USA in 2008. This indicated that the AH-PRRS20178-1 strain belonged to type 2 PRRSV. Each ORF from the AH-PRRS20178-1 genome was compared with the strain NADC30. The result showed that AH-PRRS20178-1 shared a homology of 93.51%-97.30% with the strain NADC30. In comparison with the strains NADC30 and VR-2332, two deletions were identified in the ORF1a of AH-PRRS20178-1 and the strain NADC30. The first deletion in the nsp2 coding region at position 322-432 was 111 aa long. The second deletion was also found in the nsp2 coding region, at positions 504-522 which was 19 aa long. In addition, AH-PRRS20178-1 possessed one unique aa deletion in the nsp2 coding region at position 483, different from the strain NADC30 (data not shown).

The phylogenetic tree was constructed based on the ORF5 coding sequence of PRRSVs (Figure. 3b) showed that AH-PRRS20178-1 clustered with other strains detected in China and USA formed a clade and to lineage 1 of type 2 PRRSV.”

Lines 202-206: Please consider replacing it with: “Figure 3. Porcine reproductive and respiratory syndrome virus isolated from sick pigs: (a) genomic organization of AH-PRRS20178-1, including the ORFs and the viral proteins encoding sequences; (b) phylogenetic analysis based on the ORF5 sequences of AH-PRRS20178-1 and reference strains of another type 2 PRRSVs. AH-PRRS20178-1 identified in this study is marked with a solid red circle.”

Lines 207-247: Please consider replacing it with: “In the present study, porcine circovirus type 2 (PCV2) was detected in 18 libraries (number of sequence reads matched to PCV2≥10), ten complete genomes of PCV2 were obtained from 10 different libraries and were named AH-PCV20178-1 to AH-PCV20178-10. All ten complete PCV2 genomes were 1,767 bp in length and had four ORFs (except for AH-PCV20178-1, which missed an ORF4 because of one nucleotide mutation at the start codon) (Figure. S2). ORF1 encodes a 314 aa Rep protein involved in viral replication, ORF2 encodes a 233 or 234 aa capsid protein, ORF3 encodes a 104 aa protein involved in viral pathogenesis, and ORF4 encodes a 59 aa protein associated with apoptosis suppression.

The aa identity of Rep proteins among the ten isolates (AH-PCV20178-1 to AH-PCV20178-10) was 99.0%~100% (Figure. 4a). AH-PCV20178-3 shared aa identity of 100% to AH-PCV20178-4, and AH-PCV20178-6 to AH-PCV20178-10. AH-PCV20178-1 had the highest aa identity of 99.7% to AH-PCV20178-5, while AH-PCV20178-2 shared aa identity of 99.7% to seven strains except for the strains AH- PCV20178-1 (99.0% aa identity) and AH-PCV20178-5 (99.4% aa identity). Sequence comparison based on Rep proteins of all ten strains showed high homology among them. Only a few amino acid sites were different (Figure. S3).

The aa identity of capsid proteins among the ten isolates ranged between 94.0% and 100% (Figure. 4b). AH-PCV20178-1 shared the highest aa identity of 100% to AH-PCV20178-5 based on capsid proteins. AH-PCV20178-2, AH-PCV20178-9, AH-PCV20178-3, and AH-PCV20178-10 had 100% aa identity. AH-PCV20178-6, AH-PCV20178-7, and AH-PCV20178-8 had 100% aa identity. AH-PCV20178-4 had a 99.6%  aa identity to AH-PCV20178-6, AH-PCV20178-7, and AH-PCV20178-8. Amino acid sequence comparison based on capsid proteins of all ten strains showed that four of the ten isolates had a typical "SNPLTV" motif, present in most PCV2d, and the rest had a typical "SNPRSV" motif, present in PCV2b (Figure. 5). The aa sequences of AH-PCV20178-1 to AH-PCV20178-3, and AH-PCV20178-5 were identical except for site 169, but 11 aa sites were different with the rest of six strains. In addition, an extended lysine residue presented at the C-terminal of AH-PCV20178-1 to AH-PCV20178-3 and AH-PCV20178-5, not found in the other six isolates. Noteworthy, AH-PCV20178-9 and AH-PCV20178-10 have three sites different from the remaining eight isolates, where two sites were identical to the PCV2d reference strain (GenBank FJ644927) isolated from the Zhejiang province of China. Using the complete capsid encoding sequences of these, their closest viral relatives are based on the best BLASTx hits and other representative members of PCV2s. Results from the phylogenetic tree indicated that the ten viruses fell into two distinct genotypes, including PCV2b and PCV2d (Figure. 4c). Six strains (AH-PCV20178-4 and AH-PCV20178-6 to AH-PCV20178-10) belonged to the PCV2b genotype, clustered with strains detected in China, United Kingdom, Slovakia, South Korea, South Africa, and Namibia. Four isolates (AH-PCV20178-1 to AH-PCV20178-3, and AH-PCV20178-5) belonged to the PCV2d genotype, clustered with one strain (Gene Bank FJ644927) isolated from Zhejiang of China. “

Lines 249-253: Please consider replacing it with: “Figure 4. Porcine circovirus type 2 identified in the sick pigs: (a), (b) The sequence comparison based on the amino acid sequences of Rep or capsid protein of ten PCV2 isolates identified in this study; (c) phylogenetic analysis based on the nucleotide sequences of the capsid of PCV2 identified in this study and reference strains of another PCV2s. PCV2 identified in this study are marked with a solid red circle.”

Lines 255-257: Please consider replacing it with: “Figure 5. The amino acid sequences alignment of the capsid protein of PCV2 isolated from sick pigs. The 255 differential amino acids are shown. The conservative motif "SNPRSV" was marked with a red wireframe.”

Lines 260-294: Please consider replacing it with:” Porcine parvoviruses (PPV) were detected in 39 libraries, being 20 nearly complete genomes obtained from 17 libraries. Four of the viruses belonged to PPV6 (AH-PPV620178-1 to AH- 262 PPV620178-4; ranging from 6,287 to 6,571 bp), three to PPV5 (AH-PPV520178-1 to AH-PPV520178-3; ranging from 5,480 to 5,760 bp), seven to PPV2 (AH-PPV220178-1 to 264 AH-PPV220178-7; ranging from 5,381 to 5,754 bp), five to PPV7 (AH-PPV720178- 265 1 to AH-PPV720178-5; ranging from 4,003 to 4,331 bp), and one to PPV3 (AH-PPV320178; 4,966 bp).

The genomes encoded two ORFs, except for AH-PPV320178, which has three ORFs (Figure. S4). The ORF1 encoded a nonstructural protein (NS) with a length of 661 aa for PPV2 (except AH-PPV220178-7 that was missing 12 aa at site 217 to 228), 636 aa for PPV3, 601 aa for PPV5, 662 aa for PPV6, and 672 aa for PPV7. The ORF2 encoded a structural protein (VP) with a length of 1,032 aa for 273 PPV2, 925 aa for PPV3, 991 aa for PPV5, 1,189 aa for PPV6, and 474 aa for PPV7 except for 274 AH-PPV720178-4 which missed five aa at site 181 to 185. In addition, PPV3 had an ORF3 which encoded a 555 aa major structural protein that is completely included in ORF2. The aa identity of the NS was 96.4% to 99.5% among the PPV2 strains, 99.8% to 100% among the PPV5 strains, 100% among the PPV6 strains, and 93.3% to 99.6% among the PPV7 strains. The aa identity of the VP was 94.8% to 99.7% among the PPV2 strains, 98.9% to 99.6% among the PPV5 strains, 96.6% to 99.3% among the PPV6 strains, and 91.5% to 100% among the PPV7 strains (Figure. 6). The only PPV3 strain identified shared a homology of 99.47% with the strain HK5 (GenBank EU200675) isolated from Hong Kong (China). The phylogenetic tree constructed based on the NS1 sequences indicated that the 20 viruses belonged to five different genotypes: PPV2, PPV3, PPV5, PPV6, and PPV7 (Figure. 7). Among them, seven isolates belonged to PPV2 and clustered with a strain isolated from China formed a clade. One isolate clustered with the HK5 strain isolated from China to form a clade of PPV3. Three isolates belonged to PPV5 and clustered with strains isolated from South Korea, Poland, and the United States to form a clade. Four isolates belonged to PPV6 and clustered with strains isolated from China, Brazil, Poland, and South Korea to form a clade. Five PPV7 isolates clustered with a strain isolated from China and Colombia to form a clade.”

Lines 296-299: Please consider replacing it with: ”Figure 6. Porcine parvovirus isolated from the sick pigs. Pairwise comparison based on the aa of NS protein (a, c, e, g) and VP protein of five species of PPV (b, d, f, h).

Lines 301-304: Please consider replacing it with: ”Figure 7. Please consider replacing it with: ” Figure 7. Phylogenetic analysis of PPVs isolated from sick pigs. The phylogenetic analysis was based on the NS1 sequences, including reference strains from the genus Copiparvovirus, Tetrapavovirus, Protoparvovirus, Bocaparvovirus, and Chapparvovirus. PPVs identified in the study are marked with a solid red circle.

Lines 306-312: Please consider replacing it with: ”Apart from described viruses, some Anelloviruses and Astroviruses were also identified in the pig tissue samples. Anelloviruses were present in 6 libraries, and these sequences showed high sequence identity (95%) to other Torque teno sus viruses (TTSuV). Astroviruses were present in 4 libraries, and these sequences showed high sequence identity (99%) to the strain PoAstV_VIRES_GZ01_C1 detected in the Jilin province (China). Because no large contig was obtained using the program of de novo assemble in Geneious, the phylogenetic analysis for Anellovirus and Astrovirus was not performed.

Lines 314-495: Please consider replacing it with: A variety of viruses can infect pigs and cause diseases. Zoonotic viruses from pigs can be transmitted to humans, so monitoring viruses in pigs is essential for ensuring the development of the pig industry while protecting human health. In previous studies, viral metagenomics has been applied to searching for novel viruses in various samples from pigs. These novel viruses are bocaviruses, Torque Teno, astroviruses, rotaviruses, and kobuviruses [14]. Viral metagenomics has also been widely used to explore the etiology of porcine diseases [i.e., piglet diarrhea, postweaning multisystemic wasting syndrome, porcine respiratory diseases, periweaning failure-to-thrive syndrome (PFTS)] [15,16]. Only a few viral metagenome studies have explored the etiology of porcine diseases using pig tissues. Mikael and co-workers [ref] analyzed the virus composition of tonsils from conventional pigs with lesions in the respiratory and from specific pathogen-free pigs. The authors testified that no differences were observed among the different groups. Giovanni and co-workers [ref] investigated the potential role of viral agents in PFTS-infected and healthy pigs, evaluating the virome composition of different organs. These authors demonstrated a higher abundance of porcine parvovirus 6 in healthy pigs. Contrary, Ungulate bocaparvovirus 2, Ungulate protoparvovirus 1, and porcine circovirus 3 abundance was higher in pigs with PFTS [9,16].

In the present study, we performed high-throughput sequencing to uncover the virome of various tissues collected from diseased pigs. No zoonotic viruses were detected in the tissue libraries. However, many viruses associated with pig disease were detected.

The classical swine fever virus belongs to the genus Pestivirus within the family Flaviviridae. The viral genome is a single-stranded, positive-sense RNA genome of approximately 12.3 kb with one ORF flanked by 5' and 3' untranslated regions (UTRs). The ORF encodes a 3898-aa polyprotein, later cleaved into four structural proteins (C, Erns, E1, and E2) and eight non-structural proteins (Npro, P7, NS2, NS3, NS4A, NS4B, NS5A, and NS5B). More recently, based on the phylogenetic analysis of the sequences of 53 complete E2 envelope glycoprotein gene, CSFVs were divided into five genotypes (1 to 5) and 17 subgenotypes (1.1-1.7, 2.1-2.7, 3, 4, and 5) [17]. In China, four subgenotypes (1.1, 2.1, 2.2, and 2.3) of CSFV strains 342 have been identified in mainland China and contributed to CSFV outbreaks. Among them, subgenotype 2.1 isolates, especially 2.1b, have become predominant in the last decade and endemic in many regions of China [18]. This study obtained nine complete genomes of CSFV from different porcine tissue samples. Phylogenetic analysis based on the E2 gene showed that eight isolates belonged to subgenotype 2.1, phylogenetically related to the BJ2-2017 strain isolated from Beijing (China). Another strain was assigned to subgenotype 2.5 and phylogenetically related to the GXRX2-2018 strain isolated from Guangdong (China) in 2018. The result indicated that two different subgenotypes of CSFV were endemic in this region. These results agree with previous studies that reported the subgenotype 2.1 as predominant in China.

The porcine reproductive and respiratory syndrome, characterized by reproductive failure in sows and respiratory disease in pigs of all ages, is one of the most devastating swine diseases worldwide [19]. The etiologic agent is the porcine reproductive and respiratory syndrome virus that belongs to the family Arteriviridae, order Nidovirales. This is a kind of enveloped, positive single-stranded RNA virus. The genome of PRRSV is approximately 15 kbp, and has at least 10 ORFs flanked by 5' and 3' UTRs. ORF1a and ORF1b encode viral non-structural proteins, while ORF2a, ORF2b, and ORF3-7 encode viral structural proteins [20]. PRRSV is divided into European genotype 1 and North American genotype 2, with Lelystad and VR-2332 as prototypical strains. In China, the coexistence of the two genotypes has been previously identified, but genotype 2 of PRRSV is the predominant strain [21]. In recent years, NADC30-like PRRSV has been identified in many regions of China and has caused substantial economic losses to the porcine industry. The NADC30-like PRRSV is genetically similar to the NADC30 strain, a type 2 PRRSV isolated in the United States in 2008 [22]. This study identified PRRSV in 15 porcine tissue libraries, and one nearly complete genome of PRRSV (AH-PRRS20178-1) was obtained. The AH-PRRS20178-1 strain shared the highest homology of 94.05% with the strain NADC30. Similar to the previously reported NADC30-like PRRSVs, the AH-PRRS20178-1 has 131-aa discontinuous deletions in the nsp2, including 111-aa deletion at position 322-432, 1-aa deletion at position 483, and a 19-aa deletion at position 504-522 which could distinguish themselves from other PRRSV strains. This is the first identification of NADC30-like PRRSV in Anhui Province of China. The nsp2 gene of PRRSV is highly variable and includes naturally occurring mutations, insertions, and deletions, which might be the most important marker for monitoring genetic variation and evolution of PRRSV [23]. Similar deletions have been detected in a previous study by Zhou and co-workers that reported a deletion in nsp2 was associated with milder virulence of the HP-PRRSV strains. However, they concluded that the 131-aa deletion was not related to the virulence of PRRSV in China [24].

In this study, the clinical symptoms of pigs infected with NADC30-like PRRSV were diarrhea, respiratory symptoms, and high fever. Although NADC30-like PRRSV is not as pathogenic as other highly pathogenic PRRSV, they are distinguished by a high incidence of recombination with other virus strains, which might lead to a virulence change. These characteristics probably made current vaccines ineffective and confer them much easier to escape immune surveillance.

The porcine circoviruses (PCVs), the smallest known animal viruses, belong to the genus Circovirus of the family Circoviridae. Nowadays, four PCVs have been identified, named porcine circovirus 1 (PCV1), porcine circovirus 2 (PCV2), porcine circovirus 3 (PCV3), and porcine circovirus 4 (PCV4) [25]. The present study identified PCV2 in ten libraries. The positive rate of PCV2 in this study is 20.0% (18/90) and is higher than that reported by Canal et and co-workers, that identified a 2.16% positive rate for PCV2 in porcine liver samples from slaughtered swine in the Rio Grande do Sul State, Brazil [26]. A recent survey on the prevalence of PCV2 in China from 2015 to 2019 showed that PCV2 was widely distributed throughout China. The average prevalence of PCV2 infection was 46.0% in China. The prevalence of PCV2 infection in China was 32.3% before 2015, 42.3% between 2015 and 2017, and 51.9% 396 in 2017 to 2019 [27]. This study's prevalence of PCV2 infection was lower than the average percentage in China. The prevalence of PCV2 infection may be caused by differences in sampling type or management measures implemented in each pig farm. PCV2 causes postweaning multisystemic wasting syndrome, porcine respiratory disease complex, reproductive disease, porcine dermatitis and nephropathy, and enteritis. Pigs infected with PCV2 in this study presented respiratory symptoms, diarrhea, high fever, neurological symptoms, and reproductive disorders. Based on the ORF2 gene sequences, PCV2 is divided into eight genotypes (a to h). PCV-2a, PCV-2b, and PCV-2d are widespread and similarly virulent in pigs, while the clinical significance of the remaining genotypes is unknown [1]. Recent reports indicated an ongoing genotype shift from PCV-2b to PCV-2d, and the PCV-2d has become the main epidemic strain. PCV-2d was initially called a mutant of PCV-2b and linked to potential vaccination failure cases [28]. We identified two genotypes of PCV2 here, including PCV-2b and PCV-2d. Comparing PCV-2b with PCV-2d, multiple aa sites of capsid protein changes, especially a lysine residue at the C-terminal extension of the capsid protein of PCV-2d. Recent experimental infection studies indicated that strains with a lysine residue at the C-terminal extension of the capsid protein led to increased virulence in vivo [29]. Based on the present study, we cannot confirm if PCV-2d has higher virulence than PCV-2b. Unlike the high variation of capsid protein, only a few aa sites of Rep protein were different. PCV2 generally has four open reading frames (ORFs). The proteins encoded by ORF1 to ORF3 are involved in viral replication, the immunogenic capsid protein, and the viral pathogenesis-associated protein. The ORF4 protein, as a novel discovered viral protein, induces host cell apoptosis [30]. In this study, the ORF4 is missing in the genome of the strain AH-PCV20178-1 because of the mutation of the third initiation codon from "G" to "A". Further studies need to check if the mutation can reduce the virulence of the strain AH-PCV20178-1.

Members of the family Parvoviridae, the family most commonly detected in this study, are small, non-enveloped viruses with a single-stranded DNA genome of 4-6 kb. Parvoviridae includes two subfamilies, Parvovirinae and Densovirinae, which infect vertebrates and invertebrates. The subfamily Parvovirinae is consisted of ten genera, Amdoparvovirus, Artiparvovirus, Aveparvovirus, Bocaparvovirus, Copiparvovirus, Dependoparvovirus, Erythroparvovirus, Loriparvovirus, Protoparvovirus and Tetraparvovirus [31]. Porcine parvovirus (PPV1) was a major causative agent in porcine reproductive failure, while the diseases associated with novel porcine parvoviruses (PPV2 to PPV7) have not been well characterized [32]. The present study detected five PPV species from 39 libraries, including PPV2, PPV3, PPV5, PPV6, and PPV7. Among them, 15 (16.6%) positive tissue libraries for PPV2, 3 (3.3%) positive for PPV3, 7 (7.8%) positive for PPV5, 16 (17.8%) positive for PPV6 and 12 (13.3%) positive for PPV7 were identified. The positive rate of PPVs in this study was lower in comparison to other studies by Tomasz and co-workers that identified a positive rate of 53.9% for PPV2, 15.4% for PPV3, 19.7% for PPV5, and 24.0% for PPV6 in serum samples from Polish swine farms [32]. The difference in positive rate may be caused by regional differences or differences in sampling type. In China, a previous systematic investigation showed that PPV1-7 was highly prevalent (55.4%) in nursery and finishing pigs in recent years [33]. This study's total positive PPV rate was 43.3%, lower than the previous report. To our surprise, the tissue samples were collected from diseased pigs, which showed obvious clinical symptoms, including diarrhea, respiratory syndrome, and reproductive disorder, but PPV1 as the sole causative agent of porcine parvovirus infection was not detected in this study. On the contrary, other PPVs were identified in many tissue samples. Although in the previous study, those novel PPVs (PPV2-7) have been identified in 53.9% of healthy pigs, it is not possible to rule out the possibility that they were the etiological agent of pig diseases. Further animal infection experiments need to be done to elucidate if there is any association between these novel PPVs and pig diseases. In addition, some non-pathogenic mammalian viruses such as Anelloviruses and Astroviruses were isolated from porcine tissue samples. Unsurprisingly, Anelloviruses and Astroviruses have been detected in porcine tissue samples. Berg and co-workers [ref] have previously detected those viruses in tonsil samples from conventional pigs and specific pathogen-free pigs [9]. Although almost one or more mammalian viruses have been identified from each library, some libraries did not present any mammalian viruses. It indicated that the cause of pig diseases here was not a virus but other pathogenic microbes, such as bacteria, or even physical and chemical factors. Some libraries (pig13, pig25, and pig90) only contained prokaryotic viral sequences, indirectly indicating that bacteria were the causative agent of these pig diseases. Our hypothesis about the bacterial disease in pigs was based on discovering Pseudomonas and Streptococcus phages in these libraries. Unfortunately, bacterial culture was not performed for pig13, pig25, and pig90. Streptococcus suis and pathogenic Escherichia coli were separately isolated from pig61 and pig82 by bacterial culture.

The application of viral metagenomics is becoming commonplace in humans diagnosis, prevention, and control of infectious diseases [34]. It provides novel possibilities for the direct comparative analysis of virus compositions of various clinic samples and detecting "new, emerging viruses". An excellent example of the practical applicability of viral metagenomics is the genome sequencing of SARS-CoV-2, which provides excellent assistance in preventing and controlling COVID-19 [35]. For veterinary medicine, the application of viral metagenomics is still on the road. There are considerable impediments to adopting viral metagenomics for disease diagnosis in veterinary medicine, including the sensitivity of detection, turnaround time, and cost. Firstly, the cost of viral metagenomics is more expensive than the traditional diagnostic methods. For small-scale pig farms, the viral metagenomic approach is not fit for large-scale epidemiological investigation and exploration of disease pathogens, considering the cost-benefit. Secondly, the turnaround time for metagenomics is measured in days rather than minutes as with traditional diagnostic methods. This will significantly reduce the timeliness of diagnosis and efficiency in preventing pig diseases, especially severe infectious diseases such as African swine fever. Thirdly, the sensitivity of viral metagenomics is lower than the traditional methods. The process of viral metagenomics is complex, including sample handling, library construction, sequencing, and data analysis. Error in each step will affect the quality of the library and reduce the sensitivity. In this study, we analyzed the agreement of the traditional methods (RT-PCR or RT-qPCR) and viral metagenomic approach in pig disease diagnosis. These viruses (PCV2, PRRSV, and CSFV) whose sequence reads ≥10 in the library were almost detected using traditional methods (RT-PCR or RT-qPCR), while these viruses in some samples that were detected using traditional methods had no sequence reads in the corresponding libraries. It indicated that the traditional methods had higher sensitivity than the viral metagenomic approach. Based on the shortcomings of viral metagenomics, it is now challenging to promote the diagnosis of pig disease now widely. However, through regular sampling and testing, viral metagenomics can be used to monitor the outbreak of potential diseases in pigs on those farms on a specific scale. After obtaining the associated information of the virus genome, the traditional methods will be used for further pathogen confirmation and large-scale epidemiological investigation. We believe combining viral metagenomics and traditional methods in pig disease diagnosis and prevention will have a broader prospect.

Lines 497-502: Please consider replacing it with: “This study provides an overview of the tissue virome of sick pigs and significantly increases our understanding of the prevalence status of porcine disease-associated viruses. This study provides valuable information for this area's prevention and treatment of pig disease. This study is a preliminary exploration of the application of viral metagenomics in the diagnosis of pig diseases and provides essential information for its application in the future diagnosis and prevention of pig diseases.”

Scientific comments:

Lines 81-82: please elucidate the sentence: Reads were considered duplicates if 5 to 55 were identical and only one random copy was kept. Generally, it is ”The reads were considered duplicates if bases 5 to 55 were identical”.

Author Response

The manuscript viruses-1891787, entitled “Viral metagenomics reveals diverse viruses in the tissue samples of diseased pigs”, identifies the viruses that constitute the virome in tissues collected from diseased pigs.

 We would like to thank the respected reviewer 1 for his useful comments. We have tried to consider all comments revised by the reviewer’s suggestions.

General comments:

  1. Clarity and readability of the manuscript can be improved in terms of:
  2. a)        English language
  3. b)       Typographical mistakes
  4. c)        Numbers up to eleven must be written in full
  5. d)       Consistency
  6. e)        Units from SI (µl and not ul)
  7. f)         For the first time it is used, an acronym should be explained. After, it is enough to use the acronym

Answer: Thank you for the reviewer's suggestion. We have checked carefully and revised throughout the manuscript. We believed that the clarity and readability of the manuscript have been improved.

  1. Much of the text from the result section can be converted into tables, reducing the length of the manuscript and simplifying result interpretation.

Answer: Thank you for the reviewer's suggestion. We have also considered displaying some experimental results in the form of tables, especially about virus genome length and sequence homology. However, there are so many virus data involved that it is difficult to put them in one table without confusion. In addition, even if the data is presented in the form of tables, we still need to describe and analyze it in the text. 

Specific comments:

Line 10: Please consider replacing it with “Abstract: The swine industry plays an essential role in agricultural production in China. Diseases, es-,”.

Answer: Thank you very much. We have revised it acctextording to your suggestion.

Lines 11-12: Please consider replacing with “diseases, affect the development of the pig industry and threaten human health.”.

Answer: Ok, we have revised it.

Lines 14 – 17: Please consider replacing it with “The eukaryotic viruses identified belonged to the families Anelloviridae, Arteriviridae, Astroviridae, Flaviviridae, Circoviridae, and Parvoviridae. The prokaryotic virus families, including Siphoviridae, Myoviridae, and Podoviridae, occupied a large proportion of some samples.”

Answer: Ok, we have revised it.

Lines 17 – 19: Please consider replacing it with “This study provides valuable information for understanding the tissue virome in sick pigs and for the monitoring, preventing, and treating of viral diseases in pigs.”

Answer: Ok, we have revised it.

Lines 24-27: Please consider replacing it with “China is one of the biggest pig breeding countries in the world. According to the Ministry of Agriculture and Rural Affairs of the People's Republic of China's official statistics, the number of pigs slaughtered in 2021 achieved 1.672 billion (http://www.moa.gov.cn/).”

Answer: Ok, we have revised it.

Lines 27-28: Please consider replacing it with “With the development of the economythe demands for a good environment and public health are increasing. ” This sentence seems quite decontextualized, please consider removing it.

Answer: Ok, we have revised it.

Lines 28-29: Please consider replacing it with “In China, the traditional free-range pig breeding has been changing to intensive and industrialized pig farming.”

Answer: Ok, we have revised it.

Lines 29-31: Please consider replacing it with “Highly intensive breeding poses a significant challenge for controlling and preventing pig diseases, especially viral ones, which are still the main threat to the pig industry.”

Answer: Ok, we have revised it.

Lines 31-38: Please consider replacing it with “Domesticated pigs act as reservoirs for many emerging and re-emerging viruses (e.g., classical swine fever virus, pseudorabies virus, swine influenza virus, porcine reproductive and respiratory syndrome virus, porcine circovirus, porcine parvovirus) [1–3]. Some are zoonotic viruses that recognize pigs as hosts and are transmitted to humans (e.g., hepatitis E, Nipah, influenza A, and Japanese encephalitis viruses) [4–7]. To ensure the food safety of pork and minimize the severe consequences of pig-associated zoonotic viruses to human health, the study of pig virome is of critical importance.

Answer: Ok, we have revised it.

Lines 39-41: Please consider replacing it with “Viral metagenomics is a powerful tool for exploring new and existing viruses used to elucidate the pig virome of various samples [8,9]. However, little is known about the virome in the tissue samples from diseased pigs in China.”

Answer: Ok, we have revised it.

Lines 43-44: Please consider replacing it with “In this study, we described the virome in tissue samples from diseased pigs collected on farms in China's Anhui, Jiangsu, and Sichuan provinces.”

Answer: Ok, we have revised it.

Lines 44-45: Please consider removing the sentence “Many DNA or RNA viruses associated with pig disease were identified.”

Answer: Ok, we have revised it.

Lines 45-47: Please consider replacing it with “The results draw a landscape of virome in diseased pigs and provide valuable information for the prevention and treatment of pig viral diseases.”

Answer: Ok, we have revised it.

Lines 50-58: Please consider replacing it with “From 2017 to 2018, 90 stillborn or sick pigs of different ages were sent to the Anhui Agricultural University for pathogen detection (81 pigs from the Anhui province, six from the Jiangsu province, and three from the Sichuan province) (Figure S1 and Table S1). The clinical symptoms of sick pigs included respiratory symptoms, reproductive disorders, high fever, diarrhea, weight loss, acute death, and neurological symptoms. Sick pigs were slaughtered, and depending on the clinical symptoms of different pigs, various tissues (i.e., liver, lung, lymph node, spleen, kidney, and brain) were aseptically collected. Tissue samples were sent to the laboratory on dry ice and stored at -80℃ before use.

Answer: Ok, we have revised it.

Lines 60-63: Please consider replacing it with “Each tissue sample (~25 mg) was added 800 µl phosphate-buffered saline (PBS) and homogenized in a sterilized centrifugal tube using tissue homogenizer. The tissue homogenate was frozen in a -80 â„ƒ refrigerator, thawed three times on ice and centrifuged (10 min, 15,000 ×g). The supernatant was collected.

Answer: Ok, we have revised it.

Lines 63-70: Confusing sentence, please rewrite: “Totally 500 µl tissue suspensions equal from various tissues from each pig were mixed and filtered through a 0.45-μm filter (Merck Millipore, MA, USA), to remove bacterial and eukaryotic cell-sized particles. Samples were treated with DNase (Turbo DNase from Ambion, Thermo Fisher, Waltham, MA, USA; Baseline-ZERO from Epicentre, Charlotte, USA) and RNase (Promega, Madison, WI, USA), to digest unprotected nucleic acid, at 37 â„ƒ, for 90 min. Viral RNA and DNA were extracted using the QIAamp viral RNA Minikit (Qiagen, HQ, Germany).”.

Answer: Ok, we have rewrite the sentence to “Totally 500 µl tissue suspensions were filtered through a 0.45-μm filter (Merck Millipore, MA, USA)”. Please see page 2, line 20-21.

Lines 72–75: Please consider replacing it with: The ninety viral nucleic acid pools containing DNA and RNA viral sequences were subjected to RT reactions with SuperScript III reverse transcriptase (Invitrogen, CA, USA) using (?) 100 pmol of a random hexamer primer. Klenow enzyme (New England Biolabs, MA, USA) was used to generate the complementary chain of cDNA.

Answer: Ok, we have revised it.

Lines 84-85: Please consider replacing it with: “Adaptors were trimmed using the default parameters of VecScreen, an NCBI BLASTn with specialized parameters designed for adaptor removal.”

Answer: Ok, we have revised it.

Lines 85-87: Please consider replacing it with: “Bacterial reads were subtracted by mapping the bacterial nucleotide sequences from the BLAST NT database using Bowtie2 v2.2.4.”

Answer: Ok, we have revised it.

Lines 88-90: Please consider replacing it with: “The assembled contigs and singlets were aligned to an in-house viral proteome database using BLASTx (v.2.2.7) with an E-value cut off of <10−5 .”

Answer: Ok, we have revised it.

Lines 90-93: Please consider replacing it with: “This database was compiled using the NCBI virus reference proteome (https://ftp.ncbi.nih.gov/refseq/release/viral/) and added viral protein sequences from the NCBI nr fasta file (based on annotation taxonomy in the Virus Kingdom).”

Answer: Ok, we have revised it.

Lines 93-97: Please consider replacing it with: “The candidate viral hits were compared to an in-house non-virus non-redundant (NVNR) protein database to remove false positive viral hits. The NVNR database was compiled using non-viral protein sequences extracted from the NCBI nr fasta file (based on annotation taxonomy excluding the Virus Kingdom).”

Answer: Ok, we have revised it.

Lines 97-98: Please consider replacing it with: “To obtain complete genomes or longer contigs, each viral contig was used as a reference for mapping the raw data using the Low Sensitivity/Fastest parameter in Geneious v11.1.2.”

Answer: Ok, we have revised it.

Lines 101-103: Please consider replacing it with: “Phylogenetic analyses were performed based on the predicted amino acid sequences, the closest viral relatives based on the best BLASTx hit, and the representative member of related viral species or genera.”

Answer: Ok, we have revised it.

Lines 101-103: Please consider replacing it with: “The viral genome sequences were deposited in the GenBank with the accession numbers: MW853923 to MW853962. The raw sequence reads from the metagenomic library were deposited in the Shirt Read Archive of the GenBank database (Table. S2).”

Answer: Ok, we have revised it.

Lines 116-117: Please consider replacing it with: “After next-generation sequencing with the Illumina Miseq platform, the 90 libraries generated 16,715,303 raw sequence reads.”

Answer: Ok, we have revised it.

Lines 117-119: Please consider replacing it with: “A total of 1,884,060 unique sequence reads with an Evalue cut-off of <10−5 with viral proteins (Table S2) corresponded to 11.27% of the total number of unique reads.”

Answer: Ok, we have revised it.

Lines 119-126: Please consider replacing it with: “Sequences related to prokaryotic viruses accounted for 69.94% and were affiliated with three virus families: Siphoviridae (68.92%), Myoviridae (0.76%), and Podoviridae (0.26%). Sequences related to eukaryotic viruses accounted for 30.06% and were affiliated with six virus families: Parvoviridae (24.12%), Circoviridae (3.97%), Flaviviridae (1.58%), Arteriviridae (0.3%), Anelloviridae (0.07%), and Astroviridae (0.02%) (Figure. 1a). ”

Answer: Ok, we have revised it.

Lines 127-137: Please consider replacing it with: “Figure 1b shows a heat map of each library's percentage of viral reads, catalogued into nine eukaryotic or prokaryotic viral families. The viral families that cause diseases in pigs (i.e., Parvoviridae, Circoviridae, Flaviviridae, and Arteriviridae) were distributed in different libraries. The viral sequence reads mapped to Parvoviridae had the widest distribution in libraries, followed by Circoviridae, Flaviviridae, and Arteriviridae. Only a few libraries contained viral sequence reads belonging to the family Anelloviridae, which has no association with pig diseases, and Astroviridae, which cause subclinical symptoms in infected pigs. In some libraries (i.e., pig13, pig25, pig46, pig61, pig82, and pig90) the dominant proportion of viral reads were mapped to the prokaryotic viral families of Siphoviridae, Myoviridae, and Podoviridae, which may hint bacterial diseases present in sick pigs [13].”

Answer: Ok, we have revised it.

Lines 139-141: Please consider replacing it with: “Figure 1. Composition of the virome from tissues collected from sick pigs: (a) the percentage of virus sequences from different virus families and (b) the percentage of eukaryotic or prokaryotic viral families in each library.”

Answer: Ok, we have revised it.

Lines 143-160: Please consider replacing it with: “Classical swine fever virus (CSFV) was detected in 18 libraries (number of sequence reads matched to CSFV≥10). Nine nearly complete genomes of CSFV were obtained (AH-CSFV20178-1: 12,068 bp,  AH-CSFV20178-2: 12,079 bp, AH-CSFV20178-3: 12,109 bp, AH-CSFV20178-4: 12,127 bp, AH-CSFV20178-5: 12,157 bp, AH-CSFV20178-6: 12,182 bp, AH-CSFV20178-7: 12,189 bp, AH-CSFV20178-8: 12,200 bp, AH-CSFV20178-9: 12,363 bp), which had only one open reading frame (ORF) encoding a 3,898 amino acids (aa) of the polyprotein. The aa homology of polyprotein among the eight strains (AH-CSFV20178-2 to AH-CSFV20178-9) was 99.6%, while a lower aa homology (96.6%) was obtained in the strain AH-CSFV20178-1 and the other eight strains. Twelve viral proteins were predicted by comparing the CSFV submitted sequences (GenBank NC_002657), including four structural proteins (capsid protein, RNase protein, E1, and E2) and eight non-structural proteins (Npro, P7, NS2, NS3, NS4A, NS4B, NS5A, and NS5B) (Figure. 2a). Blastn search in NCBI based on the complete genome of the nine viruses showed that eight strains (AH-CSFV20178-2 to AH- 156 CSFV20178-9) shared the highest homology (98.98%-99.16%) with strain BJ2-2017 (GenBank MG387218) detected in Beijing (China), in 2017. The strain AH-CSFV20178-1 presented high homology (97.38%) with the strain GD53/2011 (GenBank KP343640), detected in Jilin (China), in 2011.”

Answer: Ok, we have revised it.

Lines 161-172: Please consider replacing it with: “The phylogenetic trees for the encoding sequences of E2 indicated that the nine viruses belong to two distinct sub-genotypes (Figure. 2b). The eight strains (AH-CSFV20178-2 to AH-CSFV20178-9) clustered with strains isolated from China, Japan, Mongolia, Korea, Viet Nam, and Lithuania that formed a separate clade and belonged to sub-genotype 2.1. In terms of genetic distance, those eight viruses were phylogenetically more related to the strain BJ2-2017 isolated from Beijing of China in 2017, while they had a relatively distant relationship with the Elsenburg strain isolated from South Africa in 2005. The strain AH-CSFV20178-1 clustered with other strains of CSFVs detected from China, Viet Nam formed a separate clade and belonged to sub-genotype 2.5. AH-CSFV20178-1 were phylogenetically more related to the strain GXRX2-2018 isolated from Guangdong in China in 2018.”

Answer: Ok, we have revised it.

Lines 174-178: Please consider replacing it with: “Figure 2Classical swine fever virus isolated from the sick pigs: (a) genomic organization of two different CSFV representative strains, including the ORFs and viral proteins encoding sequences; (b) phylogenetic analysis based on the sequences of E2 of CSFVs identified in this study and the reference strains of other CSFVs. CSFVs identified in this study are marked with a solid red circle.”

Answer: Ok, we have revised it.

Lines 180-200: Please consider replacing it with: “In this study, porcine reproductive and respiratory syndrome virus (PRRSV) was identified in 15 libraries (number of sequence reads matched to PRRSV≥10), and one nearly complete genome was assembled from library pig28 - AH-182 PRRS20178-1. The AH-PRRS20178-1 genome was 14,940 bp in length, including a 144-bp 5' end and a 121-bp 3' end sequences (Figure. 3a), ten ORFs (ORF1a, ORF1b, ORF2a, ORF2b, ORF3, ORF4, ORF5a, 185 ORF5, ORF6 and ORF7) and shares a homology of 94.05% with the strain NADC30 (GenBank JN654459) isolated from the USA in 2008. This indicated that the AH-PRRS20178-1 strain belonged to type 2 PRRSV. Each ORF from the AH-PRRS20178-1 genome was compared with the strain NADC30. The result showed that AH-PRRS20178-1 shared a homology of 93.51%-97.30% with the strain NADC30. In comparison with the strains NADC30 and VR-2332, two deletions were identified in the ORF1a of AH-PRRS20178-1 and the strain NADC30. The first deletion in the nsp2 coding region at position 322-432 was 111 aa long. The second deletion was also found in the nsp2 coding region, at positions 504-522 which was 19 aa long. In addition, AH-PRRS20178-1 possessed one unique aa deletion in the nsp2 coding region at position 483, different from the strain NADC30 (data not shown).

The phylogenetic tree was constructed based on the ORF5 coding sequence of PRRSVs (Figure. 3b) showed that AH-PRRS20178-1 clustered with other strains detected in China and USA formed a clade and to lineage 1 of type 2 PRRSV.”

Answer: Ok, we have revised it.

Lines 202-206: Please consider replacing it with: “Figure 3Porcine reproductive and respiratory syndrome virus isolated from sick pigs: (a) genomic organization of AH-PRRS20178-1, including the ORFs and the viral proteins encoding sequences; (b) phylogenetic analysis based on the ORF5 sequences of AH-PRRS20178-1 and reference strains of another type 2 PRRSVs. AH-PRRS20178-1 identified in this study is marked with a solid red circle.”

Answer: Ok, we have revised it.

Lines 207-247: Please consider replacing it with: “In the present study, porcine circovirus type 2 (PCV2) was detected in 18 libraries (number of sequence reads matched to PCV2≥10), ten complete genomes of PCV2 were obtained from 10 different libraries and were named AH-PCV20178-1 to AH-PCV20178-10. All ten complete PCV2 genomes were 1,767 bp in length and had four ORFs (except for AH-PCV20178-1, which missed an ORF4 because of one nucleotide mutation at the start codon) (Figure. S2). ORF1 encodes a 314 aa Rep protein involved in viral replication, ORF2 encodes a 233 or 234 aa capsid protein, ORF3 encodes a 104 aa protein involved in viral pathogenesis, and ORF4 encodes a 59 aa protein associated with apoptosis suppression.

The aa identity of Rep proteins among the ten isolates (AH-PCV20178-1 to AH-PCV20178-10) was 99.0%~100% (Figure. 4a). AH-PCV20178-3 shared aa identity of 100% to AH-PCV20178-4, and AH-PCV20178-6 to AH-PCV20178-10. AH-PCV20178-1 had the highest aa identity of 99.7% to AH-PCV20178-5, while AH-PCV20178-2 shared aa identity of 99.7% to seven strains except for the strains AH- PCV20178-1 (99.0% aa identity) and AH-PCV20178-5 (99.4% aa identity). Sequence comparison based on Rep proteins of all ten strains showed high homology among them. Only a few amino acid sites were different (Figure. S3).

The aa identity of capsid proteins among the ten isolates ranged between 94.0% and 100% (Figure. 4b). AH-PCV20178-1 shared the highest aa identity of 100% to AH-PCV20178-5 based on capsid proteins. AH-PCV20178-2, AH-PCV20178-9, AH-PCV20178-3, and AH-PCV20178-10 had 100% aa identity. AH-PCV20178-6, AH-PCV20178-7, and AH-PCV20178-8 had 100% aa identity. AH-PCV20178-4 had a 99.6%  aa identity to AH-PCV20178-6, AH-PCV20178-7, and AH-PCV20178-8. Amino acid sequence comparison based on capsid proteins of all ten strains showed that four of the ten isolates had a typical "SNPLTV" motif, present in most PCV2d, and the rest had a typical "SNPRSV" motif, present in PCV2b (Figure. 5). The aa sequences of AH-PCV20178-1 to AH-PCV20178-3, and AH-PCV20178-5 were identical except for site 169, but 11 aa sites were different with the rest of six strains. In addition, an extended lysine residue presented at the C-terminal of AH-PCV20178-1 to AH-PCV20178-3 and AH-PCV20178-5, not found in the other six isolates. Noteworthy, AH-PCV20178-9 and AH-PCV20178-10 have three sites different from the remaining eight isolates, where two sites were identical to the PCV2d reference strain (GenBank FJ644927) isolated from the Zhejiang province of China. Using the complete capsid encoding sequences of these, their closest viral relatives are based on the best BLASTx hits and other representative members of PCV2s. Results from the phylogenetic tree indicated that the ten viruses fell into two distinct genotypes, including PCV2b and PCV2d (Figure. 4c). Six strains (AH-PCV20178-4 and AH-PCV20178-6 to AH-PCV20178-10) belonged to the PCV2b genotype, clustered with strains detected in China, United Kingdom, Slovakia, South Korea, South Africa, and Namibia. Four isolates (AH-PCV20178-1 to AH-PCV20178-3, and AH-PCV20178-5) belonged to the PCV2d genotype, clustered with one strain (Gene Bank FJ644927) isolated from Zhejiang of China.

Answer: Ok, we have revised it.

Lines 249-253: Please consider replacing it with: “Figure 4. Porcine circovirus type 2 identified in the sick pigs: (a), (b) The sequence comparison based on the amino acid sequences of Rep or capsid protein of ten PCV2 isolates identified in this study; (c) phylogenetic analysis based on the nucleotide sequences of the capsid of PCV2 identified in this study and reference strains of another PCV2s. PCV2 identified in this study are marked with a solid red circle.”

Answer: Ok, we have revised it.

Lines 255-257: Please consider replacing it with: “Figure 5. The amino acid sequences alignment of the capsid protein of PCV2 isolated from sick pigs. The 255 differential amino acids are shown. The conservative motif "SNPRSV" was marked with a red wireframe.”

Lines 260-294: Please consider replacing it with:” Porcine parvoviruses (PPV) were detected in 39 libraries, being 20 nearly complete genomes obtained from 17 libraries. Four of the viruses belonged to PPV6 (AH-PPV620178-1 to AH- 262 PPV620178-4; ranging from 6,287 to 6,571 bp), three to PPV5 (AH-PPV520178-1 to AH-PPV520178-3; ranging from 5,480 to 5,760 bp), seven to PPV2 (AH-PPV220178-1 to 264 AH-PPV220178-7; ranging from 5,381 to 5,754 bp), five to PPV7 (AH-PPV720178- 265 1 to AH-PPV720178-5; ranging from 4,003 to 4,331 bp), and one to PPV3 (AH-PPV320178; 4,966 bp).

Answer: Ok, we have revised it.

The genomes encoded two ORFs, except for AH-PPV320178, which has three ORFs (Figure. S4). The ORF1 encoded a nonstructural protein (NS) with a length of 661 aa for PPV2 (except AH-PPV220178-7 that was missing 12 aa at site 217 to 228), 636 aa for PPV3, 601 aa for PPV5, 662 aa for PPV6, and 672 aa for PPV7. The ORF2 encoded a structural protein (VP) with a length of 1,032 aa for 273 PPV2, 925 aa for PPV3, 991 aa for PPV5, 1,189 aa for PPV6, and 474 aa for PPV7 except for 274 AH-PPV720178-4 which missed five aa at site 181 to 185. In addition, PPV3 had an ORF3 which encoded a 555 aa major structural protein that is completely included in ORF2. The aa identity of the NS was 96.4% to 99.5% among the PPV2 strains, 99.8% to 100% among the PPV5 strains, 100% among the PPV6 strains, and 93.3% to 99.6% among the PPV7 strains. The aa identity of the VP was 94.8% to 99.7% among the PPV2 strains, 98.9% to 99.6% among the PPV5 strains, 96.6% to 99.3% among the PPV6 strains, and 91.5% to 100% among the PPV7 strains (Figure. 6). The only PPV3 strain identified shared a homology of 99.47% with the strain HK5 (GenBank EU200675) isolated from Hong Kong (China). The phylogenetic tree constructed based on the NS1 sequences indicated that the 20 viruses belonged to five different genotypes: PPV2, PPV3, PPV5, PPV6, and PPV7 (Figure. 7). Among them, seven isolates belonged to PPV2 and clustered with a strain isolated from China formed a clade. One isolate clustered with the HK5 strain isolated from China to form a clade of PPV3. Three isolates belonged to PPV5 and clustered with strains isolated from South Korea, Poland, and the United States to form a clade. Four isolates belonged to PPV6 and clustered with strains isolated from China, Brazil, Poland, and South Korea to form a clade. Five PPV7 isolates clustered with a strain isolated from China and Colombia to form a clade.”

Lines 296-299: Please consider replacing it with: ”Figure 6Porcine parvovirus isolated from the sick pigs. Pairwise comparison based on the aa of NS protein (a, c, e, g) and VP protein of five species of PPV (b, d, f, h).

Answer: Ok, we have revised it.

Lines 301-304: Please consider replacing it with: ”Figure 7. Please consider replacing it with: ” Figure 7Phylogenetic analysis of PPVs isolated from sick pigs. The phylogenetic analysis was based on the NS1 sequences, including reference strains from the genus Copiparvovirus, Tetrapavovirus, Protoparvovirus, Bocaparvovirus, and Chapparvovirus. PPVs identified in the study are marked with a solid red circle.

Answer: Ok, we have revised it.

Lines 306-312: Please consider replacing it with: ”Apart from described viruses, some Anelloviruses and Astroviruses were also identified in the pig tissue samples. Anelloviruses were present in 6 libraries, and these sequences showed high sequence identity (95%) to other Torque teno sus viruses (TTSuV). Astroviruses were present in 4 libraries, and these sequences showed high sequence identity (99%) to the strain PoAstV_VIRES_GZ01_C1 detected in the Jilin province (China). Because no large contig was obtained using the program of de novo assemble in Geneious, the phylogenetic analysis for Anellovirus and Astrovirus was not performed.

Answer: Ok, we have revised it.

Lines 314-495: Please consider replacing it with: A variety of viruses can infect pigs and cause diseases. Zoonotic viruses from pigs can be transmitted to humans, so monitoring viruses in pigs is essential for ensuring the development of the pig industry while protecting human health. In previous studies, viral metagenomics has been applied to searching for novel viruses in various samples from pigs. These novel viruses are bocaviruses, Torque Teno, astroviruses, rotaviruses, and kobuviruses [14]. Viral metagenomics has also been widely used to explore the etiology of porcine diseases [i.e., piglet diarrhea, postweaning multisystemic wasting syndrome, porcine respiratory diseases, periweaning failure-to-thrive syndrome (PFTS)] [15,16]. Only a few viral metagenome studies have explored the etiology of porcine diseases using pig tissues. Mikael and co-workers [ref] analyzed the virus composition of tonsils from conventional pigs with lesions in the respiratory and from specific pathogen-free pigs. The authors testified that no differences were observed among the different groups. Giovanni and co-workers [ref] investigated the potential role of viral agents in PFTS-infected and healthy pigs, evaluating the virome composition of different organs. These authors demonstrated a higher abundance of porcine parvovirus 6 in healthy pigs. Contrary, Ungulate bocaparvovirus 2, Ungulate protoparvovirus 1, and porcine circovirus 3 abundance was higher in pigs with PFTS [9,16].

In the present study, we performed high-throughput sequencing to uncover the virome of various tissues collected from diseased pigs. No zoonotic viruses were detected in the tissue libraries. However, many viruses associated with pig disease were detected.

The classical swine fever virus belongs to the genus Pestivirus within the family Flaviviridae. The viral genome is a single-stranded, positive-sense RNA genome of approximately 12.3 kb with one ORF flanked by 5' and 3' untranslated regions (UTRs). The ORF encodes a 3898-aa polyprotein, later cleaved into four structural proteins (C, Erns, E1, and E2) and eight non-structural proteins (Npro, P7, NS2, NS3, NS4A, NS4B, NS5A, and NS5B). More recently, based on the phylogenetic analysis of the sequences of 53 complete E2 envelope glycoprotein gene, CSFVs were divided into five genotypes (1 to 5) and 17 subgenotypes (1.1-1.7, 2.1-2.7, 3, 4, and 5) [17]. In China, four subgenotypes (1.1, 2.1, 2.2, and 2.3) of CSFV strains 342 have been identified in mainland China and contributed to CSFV outbreaks. Among them, subgenotype 2.1 isolates, especially 2.1b, have become predominant in the last decade and endemic in many regions of China [18]. This study obtained nine complete genomes of CSFV from different porcine tissue samples. Phylogenetic analysis based on the E2 gene showed that eight isolates belonged to subgenotype 2.1, phylogenetically related to the BJ2-2017 strain isolated from Beijing (China). Another strain was assigned to subgenotype 2.5 and phylogenetically related to the GXRX2-2018 strain isolated from Guangdong (China) in 2018. The result indicated that two different subgenotypes of CSFV were endemic in this region. These results agree with previous studies that reported the subgenotype 2.1 as predominant in China.

The porcine reproductive and respiratory syndrome, characterized by reproductive failure in sows and respiratory disease in pigs of all ages, is one of the most devastating swine diseases worldwide [19]. The etiologic agent is the porcine reproductive and respiratory syndrome virus that belongs to the family Arteriviridae, order Nidovirales. This is a kind of enveloped, positive single-stranded RNA virus. The genome of PRRSV is approximately 15 kbp, and has at least 10 ORFs flanked by 5' and 3' UTRs. ORF1a and ORF1b encode viral non-structural proteins, while ORF2a, ORF2b, and ORF3-7 encode viral structural proteins [20]. PRRSV is divided into European genotype 1 and North American genotype 2, with Lelystad and VR-2332 as prototypical strains. In China, the coexistence of the two genotypes has been previously identified, but genotype 2 of PRRSV is the predominant strain [21]. In recent years, NADC30-like PRRSV has been identified in many regions of China and has caused substantial economic losses to the porcine industry. The NADC30-like PRRSV is genetically similar to the NADC30 strain, a type 2 PRRSV isolated in the United States in 2008 [22]. This study identified PRRSV in 15 porcine tissue libraries, and one nearly complete genome of PRRSV (AH-PRRS20178-1) was obtained. The AH-PRRS20178-1 strain shared the highest homology of 94.05% with the strain NADC30. Similar to the previously reported NADC30-like PRRSVs, the AH-PRRS20178-1 has 131-aa discontinuous deletions in the nsp2, including 111-aa deletion at position 322-432, 1-aa deletion at position 483, and a 19-aa deletion at position 504-522 which could distinguish themselves from other PRRSV strains. This is the first identification of NADC30-like PRRSV in Anhui Province of China. The nsp2 gene of PRRSV is highly variable and includes naturally occurring mutations, insertions, and deletions, which might be the most important marker for monitoring genetic variation and evolution of PRRSV [23]. Similar deletions have been detected in a previous study by Zhou and co-workers that reported a deletion in nsp2 was associated with milder virulence of the HP-PRRSV strains. However, they concluded that the 131-aa deletion was not related to the virulence of PRRSV in China [24].

In this study, the clinical symptoms of pigs infected with NADC30-like PRRSV were diarrhea, respiratory symptoms, and high fever. Although NADC30-like PRRSV is not as pathogenic as other highly pathogenic PRRSV, they are distinguished by a high incidence of recombination with other virus strains, which might lead to a virulence change. These characteristics probably made current vaccines ineffective and confer them much easier to escape immune surveillance.

The porcine circoviruses (PCVs), the smallest known animal viruses, belong to the genus Circovirus of the family Circoviridae. Nowadays, four PCVs have been identified, named porcine circovirus 1 (PCV1), porcine circovirus 2 (PCV2), porcine circovirus 3 (PCV3), and porcine circovirus 4 (PCV4) [25]. The present study identified PCV2 in ten libraries. The positive rate of PCV2 in this study is 20.0% (18/90) and is higher than that reported by Canal et and co-workers, that identified a 2.16% positive rate for PCV2 in porcine liver samples from slaughtered swine in the Rio Grande do Sul State, Brazil [26]. A recent survey on the prevalence of PCV2 in China from 2015 to 2019 showed that PCV2 was widely distributed throughout China. The average prevalence of PCV2 infection was 46.0% in China. The prevalence of PCV2 infection in China was 32.3% before 2015, 42.3% between 2015 and 2017, and 51.9% 396 in 2017 to 2019 [27]. This study's prevalence of PCV2 infection was lower than the average percentage in China. The prevalence of PCV2 infection may be caused by differences in sampling type or management measures implemented in each pig farm. PCV2 causes postweaning multisystemic wasting syndrome, porcine respiratory disease complex, reproductive disease, porcine dermatitis and nephropathy, and enteritis. Pigs infected with PCV2 in this study presented respiratory symptoms, diarrhea, high fever, neurological symptoms, and reproductive disordersBased on the ORF2 gene sequences, PCV2 is divided into eight genotypes (a to h). PCV-2a, PCV-2b, and PCV-2d are widespread and similarly virulent in pigs, while the clinical significance of the remaining genotypes is unknown [1]. Recent reports indicated an ongoing genotype shift from PCV-2b to PCV-2d, and the PCV-2d has become the main epidemic strain. PCV-2d was initially called a mutant of PCV-2b and linked to potential vaccination failure cases [28]. We identified two genotypes of PCV2 here, including PCV-2b and PCV-2d. Comparing PCV-2b with PCV-2d, multiple aa sites of capsid protein changes, especially a lysine residue at the C-terminal extension of the capsid protein of PCV-2d. Recent experimental infection studies indicated that strains with a lysine residue at the C-terminal extension of the capsid protein led to increased virulence in vivo [29]. Based on the present study, we cannot confirm if PCV-2d has higher virulence than PCV-2b. Unlike the high variation of capsid protein, only a few aa sites of Rep protein were different. PCV2 generally has four open reading frames (ORFs). The proteins encoded by ORF1 to ORF3 are involved in viral replication, the immunogenic capsid protein, and the viral pathogenesis-associated protein. The ORF4 protein, as a novel discovered viral protein, induces host cell apoptosis [30]. In this study, the ORF4 is missing in the genome of the strain AH-PCV20178-1 because of the mutation of the third initiation codon from "G" to "A". Further studies need to check if the mutation can reduce the virulence of the strain AH-PCV20178-1.

Members of the family Parvoviridae, the family most commonly detected in this study, are small, non-enveloped viruses with a single-stranded DNA genome of 4-6 kb. Parvoviridae includes two subfamilies, Parvovirinae and Densovirinae, which infect vertebrates and invertebrates. The subfamily Parvovirinae is consisted of ten genera, Amdoparvovirus, Artiparvovirus, Aveparvovirus, Bocaparvovirus, Copiparvovirus, Dependoparvovirus, Erythroparvovirus, Loriparvovirus, Protoparvovirus and Tetraparvovirus [31]. Porcine parvovirus (PPV1) was a major causative agent in porcine reproductive failure, while the diseases associated with novel porcine parvoviruses (PPV2 to PPV7) have not been well characterized [32]. The present study detected five PPV species from 39 libraries, including PPV2, PPV3, PPV5, PPV6, and PPV7. Among them, 15 (16.6%) positive tissue libraries for PPV2, 3 (3.3%) positive for PPV3, 7 (7.8%) positive for PPV5, 16 (17.8%) positive for PPV6 and 12 (13.3%) positive for PPV7 were identified. The positive rate of PPVs in this study was lower in comparison to other studies by Tomasz and co-workers that identified a positive rate of 53.9% for PPV2, 15.4% for PPV3, 19.7% for PPV5, and 24.0% for PPV6 in erum samples from Polish swine farms [32]. The difference in positive rate may be caused by regional differences or differences in sampling type. In China, a previous systematic investigation showed that PPV1-7 was highly prevalent (55.4%) in nursery and finishing pigs in recent years [33]. This study's total positive PPV rate was 43.3%, lower than the previous report. To our surprise, the tissue samples were collected from diseased pigs, which showed obvious clinical symptoms, including diarrhea, respiratory syndrome, and reproductive disorder, but PPV1 as the sole causative agent of porcine parvovirus infection was not detected in this study. On the contrary, other PPVs were identified in many tissue samples. Although in the previous study, those novel PPVs (PPV2-7) have been identified in 53.9% of healthy pigs, it is not possible to rule out the possibility that they were the etiological agent of pig diseases. Further animal infection experiments need to be done to elucidate if there is any association between these novel PPVs and pig diseases. In addition, some non-pathogenic mammalian viruses such as Anelloviruses and Astroviruses were isolated from porcine tissue samples. Unsurprisingly, Anelloviruses and Astroviruses have been detected in porcine tissue samples. Berg and co-workers [ref] have previously detected those viruses in tonsil samples from conventional pigs and specific pathogen-free pigs [9]. Although almost one or more mammalian viruses have been identified from each library, some libraries did not present any mammalian viruses. It indicated that the cause of pig diseases here was not a virus but other pathogenic microbes, such as bacteria, or even physical and chemical factors. Some libraries (pig13, pig25, and pig90) only contained prokaryotic viral sequences, indirectly indicating that bacteria were the causative agent of these pig diseases. Our hypothesis about the bacterial disease in pigs was based on discovering Pseudomonas and Streptococcus phages in these libraries. Unfortunately, bacterial culture was not performed for pig13, pig25, and pig90. Streptococcus suis and pathogenic Escherichia coli were separately isolated from pig61 and pig82 by bacterial culture.

The application of viral metagenomics is becoming commonplace in humans diagnosis, prevention, and control of infectious diseases [34]. It provides novel possibilities for the direct comparative analysis of virus compositions of various clinic samples and detecting "new, emerging viruses". An excellent example of the practical applicability of viral metagenomics is the genome sequencing of SARS-CoV-2, which provides excellent assistance in preventing and controlling COVID-19 [35]. For veterinary medicine, the application of viral metagenomics is still on the road. There are considerable impediments to adopting viral metagenomics for disease diagnosis in veterinary medicine, including the sensitivity of detection, turnaround time, and cost. Firstly, the cost of viral metagenomics is more expensive than the traditional diagnostic methods. For small-scale pig farmsthe viral metagenomic approach is not fit for large-scale epidemiological investigation and exploration of disease pathogens, considering the cost-benefit. Secondly, the turnaround time for metagenomics is measured in days rather than minutes as with traditional diagnostic methods. This will significantly reduce the timeliness of diagnosis and efficiency in preventing pig diseases, especially severe infectious diseases such as African swine fever. Thirdly, the sensitivity of viral metagenomics is lower than the traditional methods. The process of viral metagenomics is complex, including sample handling, library construction, sequencing, and data analysis. Error in each step will affect the quality of the library and reduce the sensitivity. In this study, we analyzed the agreement of the traditional methods (RT-PCR or RT-qPCR) and viral metagenomic approach in pig disease diagnosis. These viruses (PCV2, PRRSV, and CSFV) whose sequence reads ≥10 in the library were almost detected using traditional methods (RT-PCR or RT-qPCR), while these viruses in some samples that were detected using traditional methods had no sequence reads in the corresponding libraries. It indicated that the traditional methods had higher sensitivity than the viral metagenomic approach. Based on the shortcomings of viral metagenomics, it is now challenging to promote the diagnosis of pig disease now widely. However, through regular sampling and testing, viral metagenomics can be used to monitor the outbreak of potential diseases in pigs on those farms on a specific scale. After obtaining the associated information of the virus genome, the traditional methods will be used for further pathogen confirmation and large-scale epidemiological investigation. We believe combining viral metagenomics and traditional methods in pig disease diagnosis and prevention will have a broader prospect.”

Answer: Ok, we have revised it.

Lines 497-502: Please consider replacing it with: “This study provides an overview of the tissue virome of sick pigs and significantly increases our understanding of the prevalence status of porcine disease-associated viruses. This study provides valuable information for this area's prevention and treatment of pig disease. This study is a preliminary exploration of the application of viral metagenomics in the diagnosis of pig diseases and provides essential information for its application in the future diagnosis and prevention of pig diseases.”

 Answer: Ok, we have revised it.

Scientific comments:

Lines 81-82: please elucidate the sentence: Reads were considered duplicates if 5 to 55 were identical and only one random copy was kept. Generally, it is ”The reads were considered duplicates if bases 5 to 55 were identical”.

Answer: Ok, we have revised it to “The sequence reads were considered duplicates if bases 5 to 55 were identical, only one random copy of duplicates was kept”. Please see page 2, line 36-37.

Reviewer 2 Report

The application of viral metagenomics is becoming commonplace in diagnosis, prevention, and control of infectious diseases in humans. In this study, the author analyzed the tissue virome of diseased pigs. They identified eukaryotic viruses, including Anelloviridae, Arteriviridae, Astroviridae, Flaviviridae, Circoviridae and Parvoviridae, and prokaryotic viruses, including Siphoviridae, Myoviridae and Podoviridae, which occupied a large proportion in some samples. This study provides valuable information for understanding the tissue virosome of sick pigs and also for monitoring, prevention and treatment of porcine viral diseases. The manuscript is well written and can be published unless minor revision.

1. The conclusion section needs to be rewrite.

2. A conclusion sentence is needed at the end of each paragraph or section.

3. high resolution figures are needed.

4. why do you focus on CSFV, PCV, PRRSV, and PPV needs to be discussed.

Author Response

Comments and Suggestions for Authors

The application of viral metagenomics is becoming commonplace in diagnosis, prevention, and control of infectious diseases in humans. In this study, the author analyzed the tissue virome of diseased pigs. They identified eukaryotic viruses, including Anelloviridae, Arteriviridae, Astroviridae, Flaviviridae, Circoviridae and Parvoviridae, and prokaryotic viruses, including Siphoviridae, Myoviridae and Podoviridae, which occupied a large proportion in some samples. This study provides valuable information for understanding the tissue virosome of sick pigs and also for monitoring, prevention and treatment of porcine viral diseases. The manuscript is well written and can be published unless minor revision.

Answer: We would like to thank the respected reviewer 2 for his useful comments. We have tried to consider all comments revised the manuscript based on the comments.

  1. The conclusion section needs to be rewrite.

Answer: We have rewritten the conclusion section. “Using virus metagenomics approach, we identified a variety of porcine disease-associated viruses in tissue samples. Our study provides an overview of the tissue virome of sick pigs and significantly increases our understanding of the prevalence status of porcine disease-associated viruses. Our study is a preliminary exploration of the application of viral metagenomics in the diagnosis of pig diseases and provides essential information for its application in the future diagnosis and prevention of pig diseases”. Please see page 16, line 36-41.

  1. A conclusion sentence is needed at the end of each paragraph or section.

Answer: The conclusion sentences have been added at the end of some paragraphs in the discussion section. Please see page 13, line 26-27, line 30-32, and line 49-50; page 14, line 31-32; page 16, line 1-2.

  1. high resolution figures are needed.

Answer: We have provided the high resolution figures. Please see figure 1, 3, 5, and 6.

  1. why do you focus on CSFV, PCV, PRRSV, and PPV needs to be discussed.

Answer: The reason why our study focus on CSFV, PCV, PRRSV, and PPV is, firstly these viruses were identified in this study; secondly, these viruses are widespread in pig farms of China and cause great economic losses to the pig industry. We have explained the reasons in the discussion section. Please see page 13, line 30-32.

Reviewer 3 Report

1- It would be helpful in the abstract if a broad definition of "sick" pigs were provided (a summary of the types of illness included).

2- In section 2.1, what ethical procedures were in place for the slaughter of the sick animals? (include and expand the statement from the end of the manuscript)

3- In section 2.1 more detail is required about the farm types and animals sampled. For example what is the range of farm sizes from which samples were obtained? What types and ages of pigs were sampled? How many pigs were sampled from each farm / how many farms were included altogether?

4- In section 2.1, what proportion of the sick pigs were exhibiting each type of symptom? Maybe this could be included in a table? This would be useful to see if there was a bias for a particular disease type.

5- In section 2.2, were the sample fresh or frozen prior to nucleic acid extraction?

6- In section 2.2, what type of tissue homogenizer was used?

7- In the discussion, there is reference to PRRSV-1 and -2 as genotypes. This is not correct as they are different virus species. Please update the nomenclature, and provide a reference accordingly.

8- In the results and discussion, it would be useful to analyse in more detail the findings in relation to the symptoms shown by the pigs.

Author Response

We would like to thank the respected reviewer 3 for his useful comments. We have tried to consider all comments revised by the reviewer’s suggestions.

1- It would be helpful in the abstract if a broad definition of "sick" pigs were provided (a summary of the types of illness included).

Answer: Ok, we have added the statement “(respiratory symptoms, reproductive disorders, high fever, diarrhea, weight loss, acute death and neurological symptoms)” in the abstract section.

2- In section 2.1, what ethical procedures were in place for the slaughter of the sick animals? (include and expand the statement from the end of the manuscript)

Answer: Ok, we have provided the ethical procedures for the slaughter of the sick animals. “The experiment was approved by the Jiangsu University, and Anhui Agricultural University Ethics Committee on the use of animals and complied with Chinese ethics laws and regulations. Sick pigs were slaughtered after death by electric shock”. Please see page 2, line 9-12.

3- In section 2.1 more detail is required about the farm types and animals sampled. For example what is the range of farm sizes from which samples were obtained? What types and ages of pigs were sampled? How many pigs were sampled from each farm / how many farms were included altogether?

Answer: We have provided the information about the farm sizes, pig ages and number of pig farms. Each pig was sampled from each farm. Please see page 2, line 3-4 and Table S1.

4- In section 2.1, what proportion of the sick pigs were exhibiting each type of symptom? Maybe this could be included in a table? This would be useful to see if there was a bias for a particular disease type.

Answer: We can only provide a general proportion of the sick pigs exhibiting each type of symptom, because some pigs showed multiple clinical symptoms. “The clinical symptoms of sick pigs included respiratory symptoms (28.9%), reproductive disorders (11.1%), high fever (3.3%), diarrhea (41.1%), weight loss (8.9%), acute death (5.6%) and neurological symptoms (2.2%)”. Please see page 2, line 7-9.

5- In section 2.2, were the sample fresh or frozen prior to nucleic acid extraction?

Answer: The samples were frozen at -80℃ prior to nucleic acid extraction. Please see page 2, line 13-14.

6- In section 2.2, what type of tissue homogenizer was used?

Answer: The high throughput tissue cryogrinder using in this study was purchased from Shanghai Bionoon Biotechnology Co., Ltd, and the product model is Bionoon-24LD. We have provided the associated information of tissue homogenizer in the manuscript. Please see page 2, line 16-17.

7- In the discussion, there is reference to PRRSV-1 and -2 as genotypes. This is not correct as they are different virus species. Please update the nomenclature, and provide a reference accordingly.

Answer: Ok, we have revised the “genotype” to “species” and provided a reference for it. Please see page 14, line 6-8.

8- In the results and discussion, it would be useful to analyse in more detail the findings in relation to the symptoms shown by the pigs.

Answer: Ok, we have added the associate contents of the symptoms shown by the pigs in the result section and analyzed them in detail in the discussion section. Please see page 5, line 2; page 7, line2-3; page 8, line 2-3; page 10, line 1-3; page 14, line 26-32, and line 48-50; page 15, line 33-36.